# p53 rapidly restructures 3D chromatin organization to trigger a transcriptional response

François Serra [1,8], Andrea Nieto-Aliseda [1,8], Lucía Fanlo-Escudero [1,8], Llorenç Rovirosa[1], Mónica Cabrera-Pasadas[1,2], Aleksey Lazarenkov[1], Blanca Urmeneta[1], Alvaro Alcalde-Merino [1], Emanuele M. Nola[1], Andrei L. Okorokov [3], Peter Fraser [4], Mariona Graupera [1,5,6], Sandra D. Castillo [1], Jose L. Sardina[1], Alfonso Valencia [2,5] & Biola M. Javierre [1,7] ✉

Activation of the p53 tumor suppressor triggers a transcriptional program to control cellular response to stress. However, the molecular mechanisms by which p53 controls gene transcription are not completely understood. Here, we uncover the critical role of spatio-temporal genome architecture in this process. We demonstrate that p53 drives direct and indirect changes in genome compartments, topologically associating domains, and DNA loops prior to one hour of its activation, which escort the p53 transcriptional program. Focusing on p53-bound enhancers, we report 340 genes directly regulated by p53 over a median distance of 116 kb, with 74% of these genes not previously identified. Finally, we showcase that p53 controls transcription of distal genes through newly formed and pre-existing enhancer-promoter loops in a cohesin dependent manner. Collectively, our findings demonstrate a previously unappreciated architectural role of p53 as regulator at distinct topological layers and provide a reliable set of new p53 direct target genes that may help designs of cancer therapies.

The *TP53* tumor suppressor gene, colloquially known as the guardian of the genome, is the most frequently altered gene in cancer. It encodes for a sequence-specific transcription factor named p53 that, after its activation via cell stress or DNA damage, triggers transcriptional activation of a myriad of target genes to ultimately facilitate distinct cellular outcomes, including cell cycle arrest, senescence or apoptosis among others[1]. However, the molecular mechanisms by which p53 controls gene transcription are not completely understood despite being one of the most studied gene in history.

Gene transcription is intimately associated with the three-dimensional (3D) packing of the chromatin within the nucleus and its alteration is closely linked to disease[2]. Chromatin folding involves three hierarchical levels. First, at megabases-scale, the genome can be segregated into the so-called A and B compartments. The A compartment represents active, accessible chromatin with a tendency to occupy a more central position in the nucleus. The B compartment corresponds to heterochromatin and is enriched at the nuclear periphery. Second, topologically associating domains (TADs) are sub-megabase structures delimited by TAD borders that interact more frequently within themselves than with the rest of the genome. TADs are conserved across species[3] and cell types and show a coordinated transcriptional status. Third, these domains are

[1]Josep Carreras Leukaemia Research Institute, Barcelona, Spain. [2]Barcelona Supercomputing Center, Barcelona, Spain. [3]Wolfson Institute for Biomedical Research, University College London, London, UK. [4]Department of Biological Science, Florida State University, Tallahassee, FL, USA. [5]Catalan Institution for Research and Advanced Studies (ICREA), Barcelona, Spain. [6]CIBERONC, Instituto de Salud Carlos III, Madrid, Spain. [7]Institute for Health Science Research Germans Trias i Pujol, Barcelona, Spain. [8]These authors contributed equally: François Serra, Andrea Nieto-Aliseda, Lucía Fanlo-Escudero. ✉e-mail: bmjavierre@carrerasresearch.org

formed by assemblies of chromatin loops[4]. Among hierarchical levels, chromatin loops engaging gene promoters and distal regulatory elements (e.g., enhancers or silencers) are key to control gene transcription since these often locate distally in the genome but need physical proximity to function[5].

Despite its critical role in gene transcription regulation, the interplay between spatio-temporal genome organization and p53 remains largely unexplored. Abrams and colleagues previously demonstrated the existence of long-range interactions between a p53-bound enhancer and multiple targets in *Drosophila*, but the chromatin configuration was unaffected by p53 status or DNA damage[6]. Similarly, Agami and colleagues studied three p53-bound enhancers that interacted with multiple distant genes to confer p53-dependent regulation in humans[7]. In addition, focused on two genes controlled by p53, Gadreau and colleagues demonstrated that transcription triggers topological changes that do not affect their connectivity with p53-bound enhancers. These studies agreed that p53-engaged DNA loops represented pre-existing and non-dynamic chromatin architectures[8]. However, since only a few p53-bound elements or p53 target genes were analyzed, a genome-wide study is needed for the generalization of these mechanistic insights. Besides, since p53 frequently binds enhancers and most of its linked distal target genes are unknown, a genome-wide study of the spatio-temporal genome organization would also represent a direct way of identifying the true set of genes directly regulated by p53.

Transcription factor CCCTC-binding factor (CTCF) and the ring-shaped multiprotein complex cohesin, which includes the major subunit RAD21, play key roles in establishing TADs and loops thought a process called ATP-dependent loop extrusion. In this process, the cohesin complex is loaded onto DNA and pushes it until encountering a barrier, which is often formed by CTCF[9–13]. Therefore, cohesin degradation leads to rapid disappearance of TADs, antagonizes compartment segregation and promotes loss of most of the significant promoter-anchored interactions over ~50Kb of length[10,13–22]. Surprisingly, complete removal of cohesin has little impact on steady-state gene transcription[10] but prevents adequate activation of inflammatory gene response[23] and initiation of new regulatory programs in neurons acquiring response to new stimuli[24]. However, the role of cohesin in activation of other types of inducible genes, such as those activated by p53, remains largely unexplored[6–8].

Here, we addressed, in a genome-wide manner for the first time, the relationship between p53 activation and spatio-temporal genome architecture to establish a transcriptional response to cellular stress. First, we uncovered a new role of p53 as a master remodeler of the spatio-temporal chromatin organization. Unexpectedly, p53-driven topological changes, including the formation of long-range DNA loops between p53-bound enhancers and promoters, occur 1 hour (h) after its activation and enable the establishment of the p53 transcriptional response. Second, we discovered an unforeseen dependence of p53 inducible gene expression on cohesin-mediated DNA looping. Finally, we identified a new set of direct target genes distally controlled by p53 via a mechanism that relies on dynamic and non-dynamic p53-bound enhancer and gene promoter interactions. Taken together, our results demonstrate the power of integrating 3D genome architecture, epigenetics, gene expression and transcription factor binding profiles along time to gain insight into gene regulatory mechanisms and to discover new transcription factor target genes.

## Results

### p53 activation drives two waves of dramatic changes in genome compartments and TADs

To address global consequences of p53 activation in 3D genome organization we used the HCT116 cell line, a widely used model characterized by a wild-type p53 response. These cells were treated with 10 μM Nutlin-3a, an inhibitor of the p53-HDM2 interaction used to mimic cell stress-induced p53 activation, and collected at five time points (1, 4, 7, 10, and 24 h post treatment). We also treated cells with the drug vehicle (i.e., DMSO) and used these cells as a control of basal conditions without p53 activation (also referred to in the manuscript as 0 h of Nutlin-3a) (Fig. 1A). First, we validated p53 activation by confirming progressive cell cycle arrest occurring in parallel with a significant p53 activation (according to phosphorylation of serine 46) and upregulation of well-known p53 target genes (i.e., *CDKN1A, TGFA, BAX*) (Suppl. Fig. 1 A–G and Suppl. Data 1, 2). We then used in situ Hi-C to generate genome-wide chromosome conformation maps along all six time points of p53 activation (Suppl. Data 3). After confirming libraries' quality and reproducibility between biological replicates (SCC > 0.95; see methods) (Suppl. Fig. 1H, I), we segmented the genome into A and B compartments. Clustering of biological replicates and separation of treatment conditions demonstrated robust and dynamic A/B compartmentalization associated with p53 activation (Fig. 1B and Suppl. Fig. 1J, K). Globally, although 87.5% of the genome compartments remained stable throughout all time points, 12.5% of A or B compartments switched during p53 activation, from here on out referred to as dynamic compartments (Fig. 1C and Suppl. Fig. 1L). Furthermore, per time points, we observed compartment activation at 1 h, limited compartment rewiring from 1 to 7 h and compartment inactivation at 10 h of p53 activation and onwards (Fig. 1D and Suppl. Fig. 1M, N), which is exemplified by a 40 Mb genomic region of chromosome 12 (Fig. 1E and Suppl. Fig. 1O). Specifically, clustering based on compartment scores identified 7 main clusters of dynamic compartments following p53 activation. Interestingly, 23.2% of these (clusters 1 and 2) dramatically flipped from B to A after 1 h of Nutlin-3a treatment and then progressively reduced their compartment score switching back to B. Contrarily, 39.5% of the dynamic compartments (clusters 3, 4, and 5) were A compartments specifically inactivated at late time points (10 or 24 h of Nutlin-3a treatment). However, these also increased their compartment score after 1 h of p53 activation despite already being classified as A compartments before p53 activation (i.e., 0 h), reinforcing the observed unexpected tendency of early, genome-wide gain of genome activity.

Next, we used chromosome-wide insulation potential (TADbit score > 4[25]) to identify 4610 unique TAD borders during p53 activation. Specifically, we detected between 2820 and 3963 TADs per time point, with median sizes ranging from 453 to 622 kilobases (Fig. 2A, B). Genome-wide insulation scores analyzed by Principal Component Analysis (PCA) and hierarchical clustering over time revealed high reproducibility between biological replicates and progressive changes reflecting p53 activation (Fig. 2C and Suppl. Fig. 2A, B). While 2032 of TAD borders were stable across all stages, the remaining ones were dynamically lost or gained. Similar to A/B compartments, 1 h of p53 activation promoted architectural change, this time resulting in loss of TAD borders (Fig. 2D, E and Suppl. Fig. 2C). TAD profiles were then extensively conserved until the second wave of rewiring occurring after 10 h of treatment. At this stage, we observed an acquisition of TAD borders followed by, after 24 h of treatment, a global erase of these topological features. Specifically, 26.3% of the dynamic TAD borders rapidly disappeared after 1 h of treatment (clusters 2 and 4, Fig. 2E). One-third of these (cluster 4) temporally reappeared after 10 h of p53 activation before disappearing again at 24 h. Moreover, 16.9% and 8.1% of the dynamic TAD borders specifically disappeared (clusters 3 and 4) or appeared (cluster 5) at 24 h of treatment, respectively. Besides these trends of appearance and disappearance, we observed a global and progressive increase in the insulation capacity of TAD borders (according to their insulation scores), suggesting that less insulating borders are preferentially erased during p53 activation (Fig. 2F and Suppl. Fig. 2D).

Finally, we tested the reversibility of these changes in genome architecture that occur during p53 activation using in situ Hi-C libraries (Suppl. Data 3). To do so, we washed Nutlin-3a after 24 h of treatment

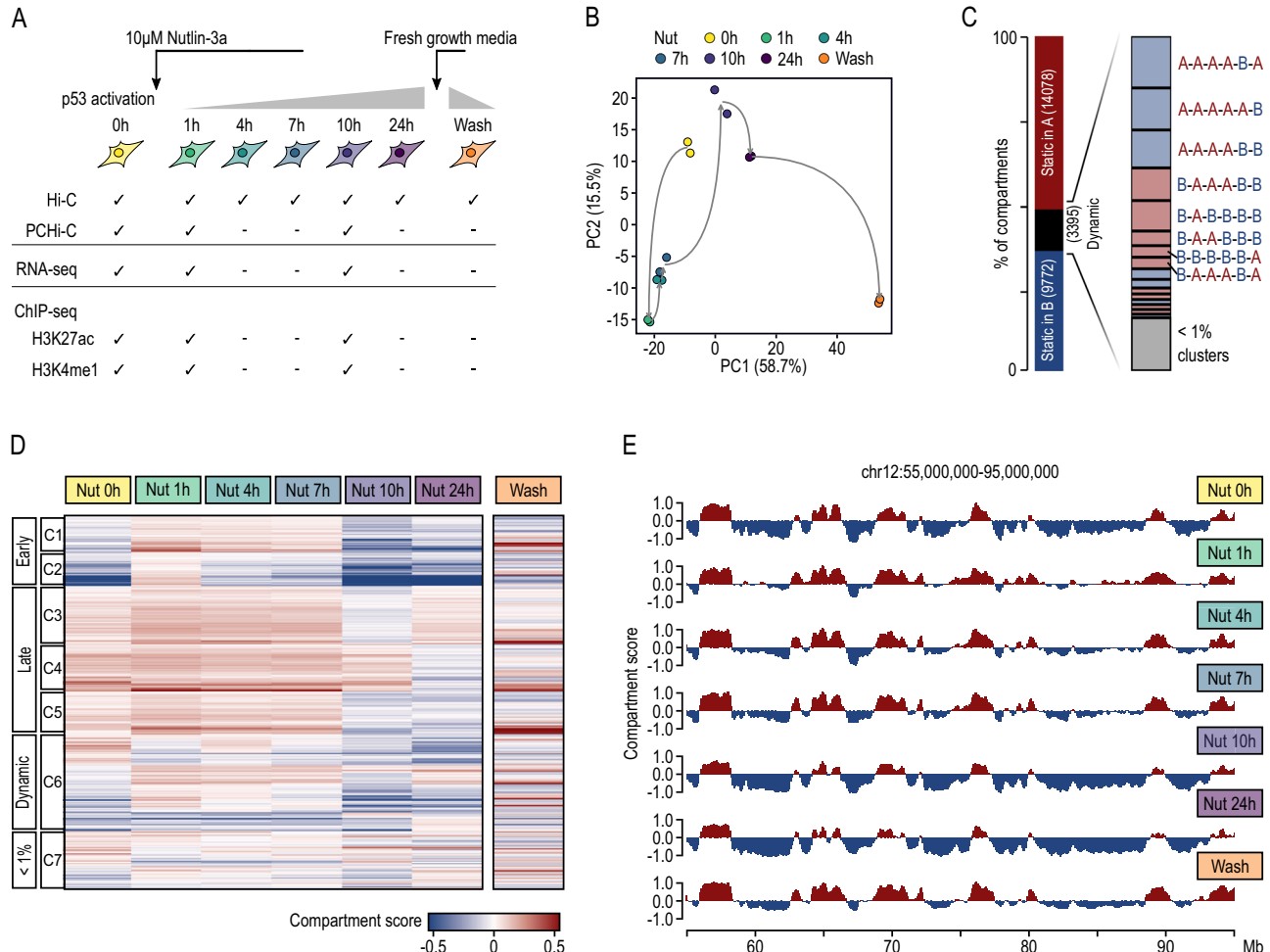

**Fig. 1 | A/B compartment changes along p53 activation. A** Experimental multiomics framework of the work. Nut 0 h means control cells treated with drug vehicle characterized by no p53 activation. Wash refers to cells treated with Nutlin-3a for 24 h, washed with PBS to remove Nutlin-3a and cultured for 24 additional hours with fresh growth media. **B** Principal Component (PC) Analysis of the compartment scores for each Hi-C individual biological replicate across p53 activation and inactivation. PC 1 and 2 were taken into consideration. Numbers in parentheses represent the percentage of variance explained by each principal component. **C** Percentage of compartment categories according to their dynamics along p53 activation. Dynamic compartments are defined as those that change compartment between any two time points. **D** Compartment scores of the dynamic compartments in response to p53 activation and inactivation, with each row representing a 100 kb bin and each column representing a time point. The heatmap color scale represents the compartment score assigned to each 100 kb bin with positive values indicating A compartment and negative values indicating B compartment. **E** Variation of PC values along a 40 Mb genomic region of chromosome 12. The genomic bins are colored according to their compartment type.

and cultured HCT116 cells for an additional 24 h in fresh growth media without Nutlin-3a (Fig. 1A). p53 protein abundance and p53 activation levels (according to phosphorylation of serin 46) of washed cells were similar to cells prior to p53 activation (Suppl. Fig. 1D–G), whereas the p53 driven transcriptional changes were not fully recovered (Suppl. Fig. 1B, C). After confirming libraries quality and reproducibility between biological replicates (SCC > 0.95, see methods) (Suppl. Fig. 1H, I), we segmented the genome into A and B compartments and identified TAD borders. Globally, we observed that a significant proportion of topological changes occurring throughout p53 activation are reversible, especially changes associated with the second wave of rewiring. This was clearly observed at the TAD level where, among the dynamic TAD borders, the percentage of shared TAD borders with respect to DMSO condition goes from 24% at 24 h to 43% in washed cells (Fig. 2 and Suppl. Fig. 2A–C) and, to lesser extent, at the compartment level (Fig. 1B, D, E and Suppl. Fig. 1J–M, O).

Altogether, these results demonstrate dramatic changes in genome architecture during p53 activation, including early changes at 1 h of activation and late changes after 10 h of treatment, that are partially reverted upon p53 inactivation.

## Early and late genome organization changes are triggered by p53 binding directly and indirectly

Since p53 activation leads to two waves of genome organization dynamics, we decided to explore the underlying molecular mechanisms driving these sharp transitions. To do so, using cells collected after 1 and 10 h of Nutlin-3a treatment and control cells treated with the drug vehicle (0 h), we generated: i) high-quality chromatin immunoprecipitation with massively parallel sequencing (ChIP-seq) libraries of two histone modifications (H3K4me1 and H3K27ac), enabling the identification of primed enhancers (H3K4me1), active enhancers (H3K4me1 and H3K27ac) and active promoters (H3K27ac) (Suppl. Data 4); and ii) RNA-seq libraries that enable the genome-wide profiling of gene transcription (Fig. 1A and Suppl. Data 5). Besides, we also used already processed publicly available ChIP-seq data of p53's binding profile in the same cell type subjected to 12 h of 10 μM Nutlin-3a treatment[26].

After verifying the libraries' quality and reproducibility between biological replicates (Suppl. Fig. 3), we identified enhancers and defined their activities along p53 activation (Suppl. Fig. 4A–D and Suppl. Data 6). Next, we further tested whether changes induced by

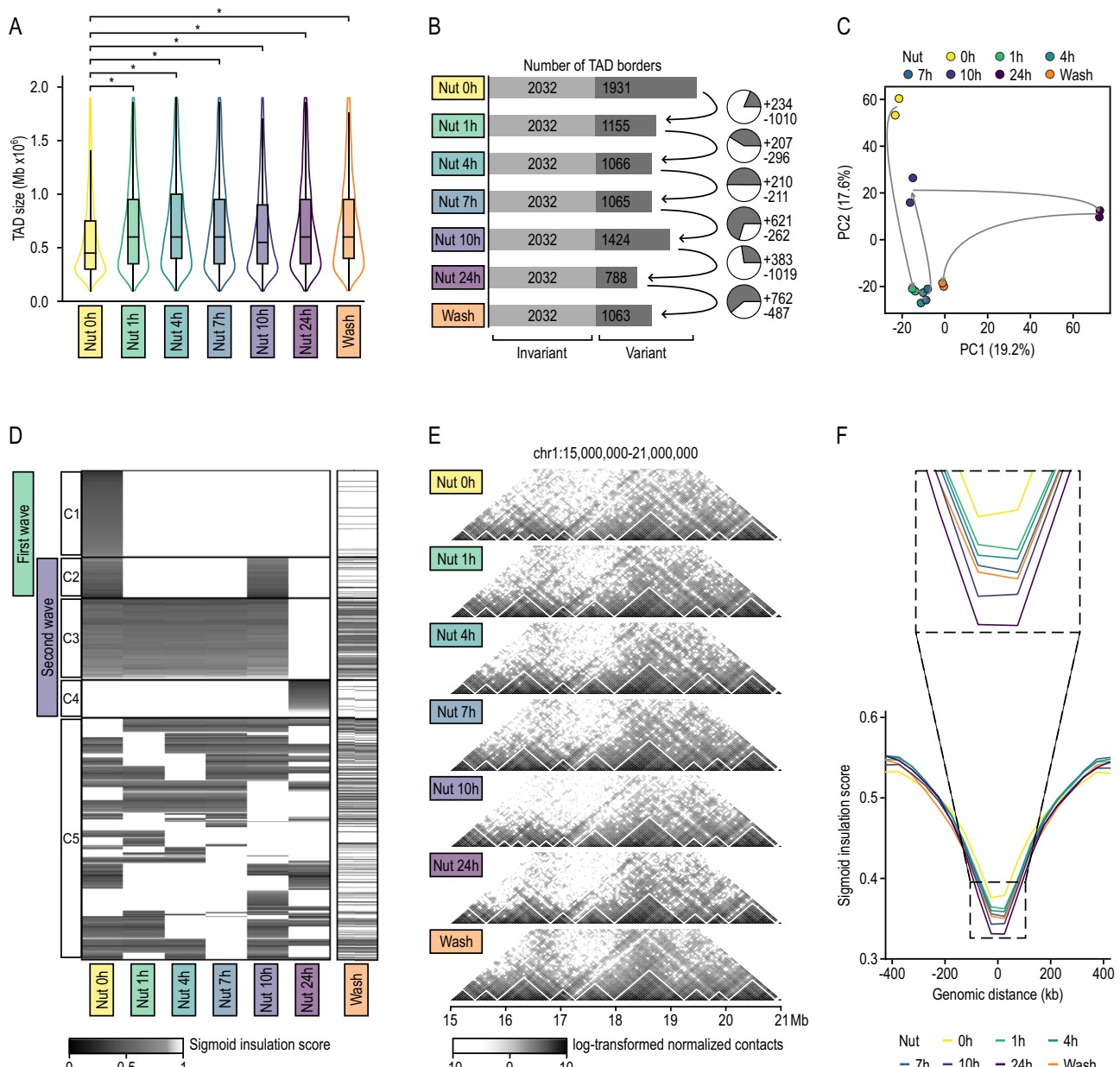

**Fig. 2 | TAD dynamics along p53 activation. A** Distribution of TAD sizes along p53 activation and inactivation. A two-sided Mann-Whitney test was used to test whether distributions of TAD sizes differed throughout p53 activation compared to control. Nut 0 h refers to control cells without p53 activation. The boxes show the interquartile range (IQR), the central line represents the median, the whiskers add 1.5 times the IQR to the 75 percentile (box upper limit) and subtract 1.5 times the IQR from the 25 percentile (box lower limit). Significance calculated with two-sided Mann–Whitney test. The star indicates *p* values below 0.01. Sample sizes of the box plots appear in the next panel. **B** Bar plot displaying the number of variant and invariant TAD borders along p53 activation and inactivation. Invariant TAD borders, in light gray, are those detected at all time points. Pie charts and positive/negative numbers represent the number of TAD borders gained/lost between the corresponding time points. **C** Principal Component (PC) Analysis of TADbit TAD border

scores for all Hi-C biological replicates across p53 activation and inactivation. Principal components 1 and 2 were taken into consideration. Numbers in parentheses represent the percentage of variance explained by each principal component. **D** Ward hierarchical clustering of TAD borders based on their TAD insulation scores and the patterns of temporal changes observed throughout p53 activation and inactivation. Only TAD borders characterized by a TADbit score >4 were included. TAD borders were manually clustered considering their cluster robustness throughout p53 activation. Cluster C5 contains all TAD borders from clusters with low support. **E** Hi-C contact maps of a 6 Mb genomic region of chromosome 1 where TADs are marked by white lines. **F** Insulation score profiles stacked over a 1 Mb genomic region centered at TAD borders. Only TAD borders characterized by a TADbit score >4 were included. Sigmoid insulation score is inversely proportional to insulation capacity.

Nutlin-3a treatment were direct (i.e., local shifts provoked by p53 binding) or indirect effects of p53 (e.g., alterations through transcription factors regulated by p53). p53 binding to chromatin is largely invariant but not all binding events deliver transactivation, leading to cell type and stimulus-specific variation in the p53 transcriptional program[27–29]. Thus, we defined as functional p53 binding sites a set of 2105 sites bound by p53 and characterized by the presence of the

activating H3K27ac histone mark in the time points corresponding to p53 activation condition (i.e., 1 or 10 h of Nutlin-3a treatment) (Suppl. Fig. 4E−G and Suppl. Data 7). Only 6.5% (137/2,105) of those functional p53 bindings occurred at promoters (i.e., −1000 + 200 bp of any TSS). Genes involved in these bindings had some degree of activity at basal conditions (i.e., H3K27ac at promoters), and tended to gain promoter H3K27ac deposition and gene transcription levels after Nutlin-3a

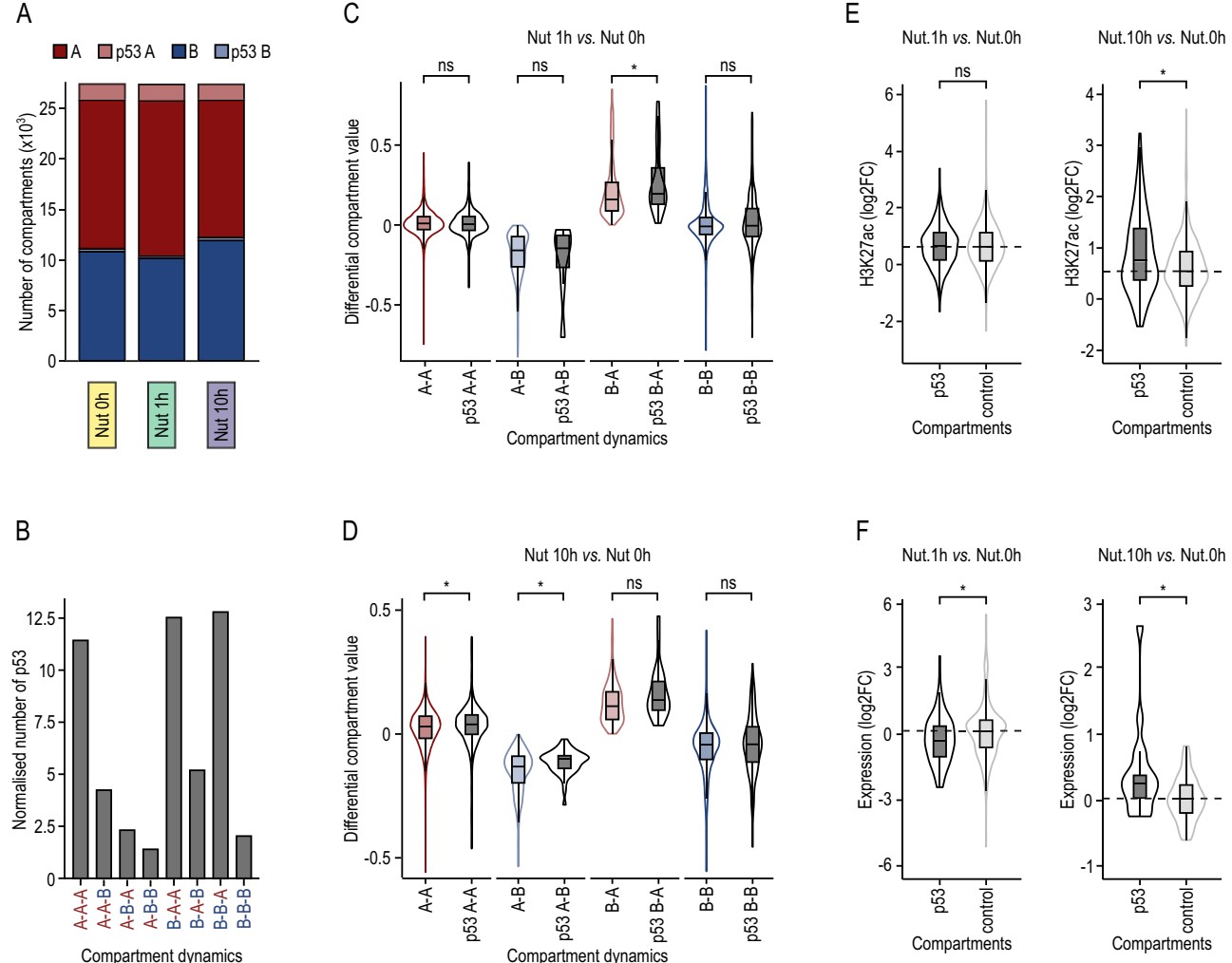

**Fig. 3 | p53 binding to chromatin leads to direct changes in 3D genome topology. A** Proportion of functional p53 bindings in A/B compartments along p53 activation. p53A/p53B are 100 kb bins in A/B compartment with at least one functional p53 binding. Nut 0 h refers to control cells without p53 activation. **B** Proportion of functional p53 bindings in the different categories of A/B compartments dynamics. Data are normalized by the total number of 100 kb regions considered in each sample. **C** Distribution of differential compartment scores between compartments defined after 1 h of p53 activation and control compartments (Nut 1 h–Nut 0 h). 100 kb regions were classified according to their compartment dynamism and according to the presence or absence of functional p53 binding sites. Two-sided Mann-Whitney test was performed to test whether differential compartment scores differed significantly between compartments with the same dynamics with and without functional p53 binding sites. The number of 100 kb bins in each category is AA: 16,050; p53-AA: 1671; AB:416, p53-AB: 7; BA:1156, p53-BA: 67; BB:10,154, p53-BB: 216. **D** As in (**C**), but in this case differential

compartments scores are between compartments defined after 10 h of p53 activation and control compartments (Nut 10 h–Nut 0 h). The number of 100 kb bins in each category is AA: 15,132; p53-AA: 1630; AB:1334, p53-AB: 48; BA:195, p53-BA: 24; BB:11,115, p53-BB: 259. **E** Differential H3K27ac between 1 h (Nut 1 h) and 0 h (Nut 0 h) time points (log2FC), identified at 100 kb regions (compartments) either with (p53) or without (control) an overlapping p53 binding site. Two-sided Mann-Whitney test was performed to test whether p53 and control distributions differed significantly. The number of 100 kb bins in each category is p53: 1961; control: 25,815. **F** As in (**E**), but in this case analyzing the differential H3K27ac between 10 h and 0 h of p53 activation. The number of 100 kb bins in each category is p53: 1961; control: 25,815. **C–F** The boxes show the interquartile range (IQR), the central line represents the median, the whiskers add 1.5 times the IQR to the 75 percentile (box upper limit) and subtract 1.5 times the IQR from the 25 percentile (box lower limit). Significance calculated with two-sided Mann–Whitney test. The star indicates *p* values below 0.05.

treatment (Suppl. Fig. 4G–I). The remaining functional p53 binding events were distal from TSS, and among them, 901 occurred at active enhancers characterized under p53 activation conditions (i.e., co-presence of H3K4me1 and H3K27ac at 1 or 10 h of Nutlin-3a treatment at regions that do not overlap promoters) (Suppl. Fig. 4G). Interestingly, most of these active enhancers were already established at basal conditions and increased their H3K27ac levels as a consequence to p53 activation (Suppl. Fig. 4G–J). Collectively, these results highlight p53's preference to bind active regulatory elements, mainly active enhancers, and suggest its limited capacity for epigenetic rewiring on the linear genome.

Since p53 preferentially acts on pre-established regulatory elements, we explored whether its functional binding may promote

topological perturbations to ultimately explain the transcriptional response. Globally, functional p53 preferentially bound A compartments (Fig. 3A) and TAD border vicinities (Suppl. Fig. 2E). Interestingly, its binding was mainly enriched in compartments that gained activity during p53 activation (Fig. 3B). In other words, compartments switching from B to A showed significant enrichment in functional p53 binding sites with respect to those switching from A to B. Indeed, after 1 h and 10 h of activation, 5.7% (61 out of 1078; Fisher O.R. 3.5 *p* value 0.007) and 13.1% (22 out of 168; Fisher OR 3.9 *p* value 4e-6) of the B to A switched compartments, respectively, were characterized by harboring functional p53 binding sites. Besides, regions harboring functional p53 binding sites (independently of the compartment dynamics) tended to suffer a higher increase in compartment score (Fig. 3C, D),

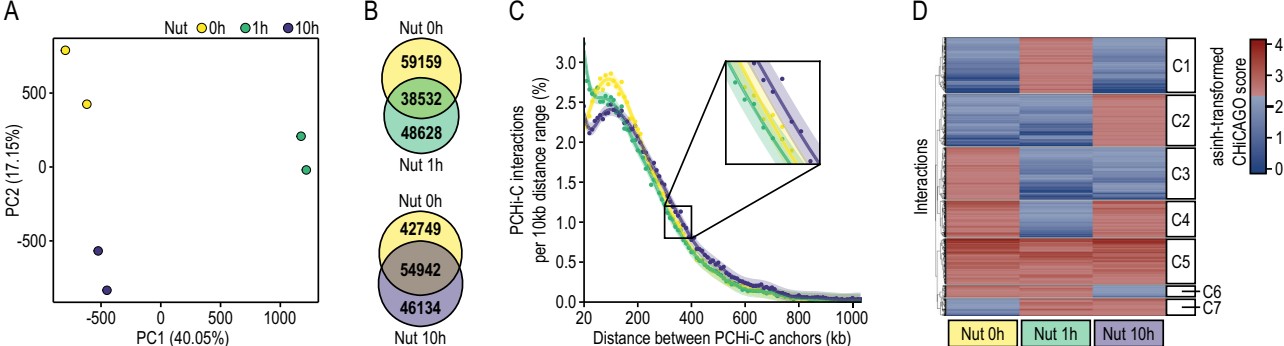

**Fig. 4 | Promoter interactions shift during p53 activation. A** Principal Component (PC) Analysis of promoter interactomes throughout p53 activation obtained by PCHi-C at the biological replicate level. Numbers in parentheses represent the percentage of variance explained by each principal component. **B** Overlap between significant promoter interactions (CHiCAGO score ≥ 5) before and after 1 h of p53 activation (top) and before and after 10 h of p53 activation (bottom). **C** Observed number of interactions with CHICAGO score ≥ 5 ranked according to the genomic distance between their anchor points. Distances were grouped into 10 kb bins and converted to percentages to facilitate the comparison between samples. Observed percentages were fitted using a linear GAM (see Methods). **D** Heatmap of asinh-transformed CHiCAGO scores. Only interactions found significant in at least one-time point were considered. Clusters were defined based on the presence or absence of interaction significance in each time point, and hierarchical clustering was performed within each predefined cluster to order the interactions order the clusters from C1 to C7. Euclidean distances were measured between interactions and clusters, and the "complete" agglomeration method was used.

a higher gain of activating H3K27ac mark (Fig. 3E), and increased transcription (Fig. 3F) compared to their counterparts depleted of functional p53 bindings. Contrarily, only 1.7% (7 out of 411) and 3.7% (48 out of 1281) of the compartments that switched from A to B after 1 or 10 h of activation, respectively, harbor functional p53 binding sites, although these tended to have a more moderate decrease in compartment score than compartments depleted of functional p53 binding (light blue area in Fig. 3C, D).

Collectively, these results show that direct p53 binding drives a displacement towards A compartments, supporting the role of p53 acting as a transcriptional activator and suggesting that indirect p53 effects, occurring at later time points, are more heterogeneous depending on the activating or repressing nature of downstream effectors (e.g., p21[30], E2F7[31], and miRNAs[32]).

### Promoter interactome is highly rewired by p53 activation

Since TADs and DNA loops are thought to share a cohesin-driven loop extrusion formation mechanism[10,15], we investigated whether our observed TAD dynamics during p53 activation may also be accompanied by promoter loop rewiring. To do so, we generate high-quality and reproducible Promoter Capture Hi-C (PCHi-C) libraries using cells collected after 1 and 10 h of Nutlin-3a treatment and control cells treated with the drug vehicle (0 h) (Fig. 1A). Each sample was deep-sequenced, paired-end reads were mapped and filtered using the HiCUP pipeline, and significant interactions of 31,253 annotated promoters were called using the CHiCAGO pipeline (CHiCAGO score ≥ 5) (Suppl. Data 8). Specifically, we detected an average of 78,832 significant interactions per sample (SD = 4067), which were characterized by a median linear distance between promoters and their interacting regions of 260 kb (SD = 12 kb) and an average of 83% (SD = 2.5%) of promoter-to-non-promoter interactions (Suppl. Fig. 5A, B). PCA of CHiCAGO interaction scores demonstrated that promoter interactomes were highly reproducible within each condition and dynamic according to p53 activation state, which was also confirmed by hierarchical clustering (Fig. 4A and Suppl. Fig. 5C, D). Consistent with our previous Hi-C data, 1 h of p53 activation led to the most dramatic changes in the promoter interactome. Specifically, early p53 activation triggered a loss of promoter interactions, mostly affecting longer interactions (>200 Kb) (Fig. 4B–C and Suppl. Fig. 5E), and an increase of promoter-promoter connectivity (Suppl. Fig. 5B). Late p53 activation (10 h of Nutlin-3a treatment) also led to specific promoter-centric topologies, which included an increase of

longer interactions (>300 Kb). A closer analysis allowed us to identify clusters of promoter interactions specifically gained at 1 h (C1), 10 h (C2) or both extensions of treatment (C7) of Nutlin-3a (Fig. 4D). Besides, we also recognized clusters of promoter interactions specifically lost during early (C3-4) and late (C6) p53 activation, some of these later re-established at 10 h of Nutlin-3a treatment (C4).

Summarizing, our data demonstrated that p53 activation leads to a dramatic rewiring of the promoter interactome that may contribute to reshaping the promoter-enhancer interactome landscape to ultimately control the transcriptional response.

### p53 controls transcription of previously non-associated genes located a few hundred kilobases away

Considering our results and previous knowledge, p53 binds preferentially to distal regulatory elements. In consequence, its target genes are often not directly identifiable, which accounts for a critical gap of knowledge. p53 regulation can occur from hundreds of kilobases away, bypassing several proximal genes, and can even occur in consequence to p53 binding in the intron of a non-target gene[33]. However, transcriptional regulation is conveyed through the physical proximity of communicating enhancers and promoters[34]. Therefore, we used our PCHi-C data to associate 253 p53-bound enhancers with 340 candidate genes to be distally (i.e., over distance) controlled by p53, from now on referred to as p53 distal target genes (Fig. 5A–B and Suppl. Data 9). Among these, we identified examples of previously identified p53 target genes (e.g., *PPM1D*[35], *TP53INP1*[36], *PLK2*[37], *DKK1*[38], *FUCA1*[39]), but also identified potential new p53 target genes[40] (e.g., *TGFR2*, *JAG2*, *BRD7*, *TENT4B*, *CD9*). p53 distal target genes were located at a median genomic distance of 116 kb from the functional p53 binding sites (Fig. 5C). Remarkably, only 7.1% (24/340) of these p53-bound enhancers were linked to the nearest gene, highlighting the importance of our long-range promoter interaction data to avoid misleading associations based on proximity within the genomic sequence. Indeed, unlike genes closest to the p53-bound enhancers along the linear genome (labeled as nearest genes in Fig. 5B), p53 distal target genes were enriched in p53-related gene sets (e.g., Reactome transcriptional regulation by *TP53* pathway[41], p53 dn.v1 up gene set[42]) (Fig. 5D and Suppl. Data 10).

Next, we validated these p53 distal target genes. First, we observed that enhancers and promoters bound by functional *TP53* binding sites gained significantly more H3K27ac than their control regions during p53

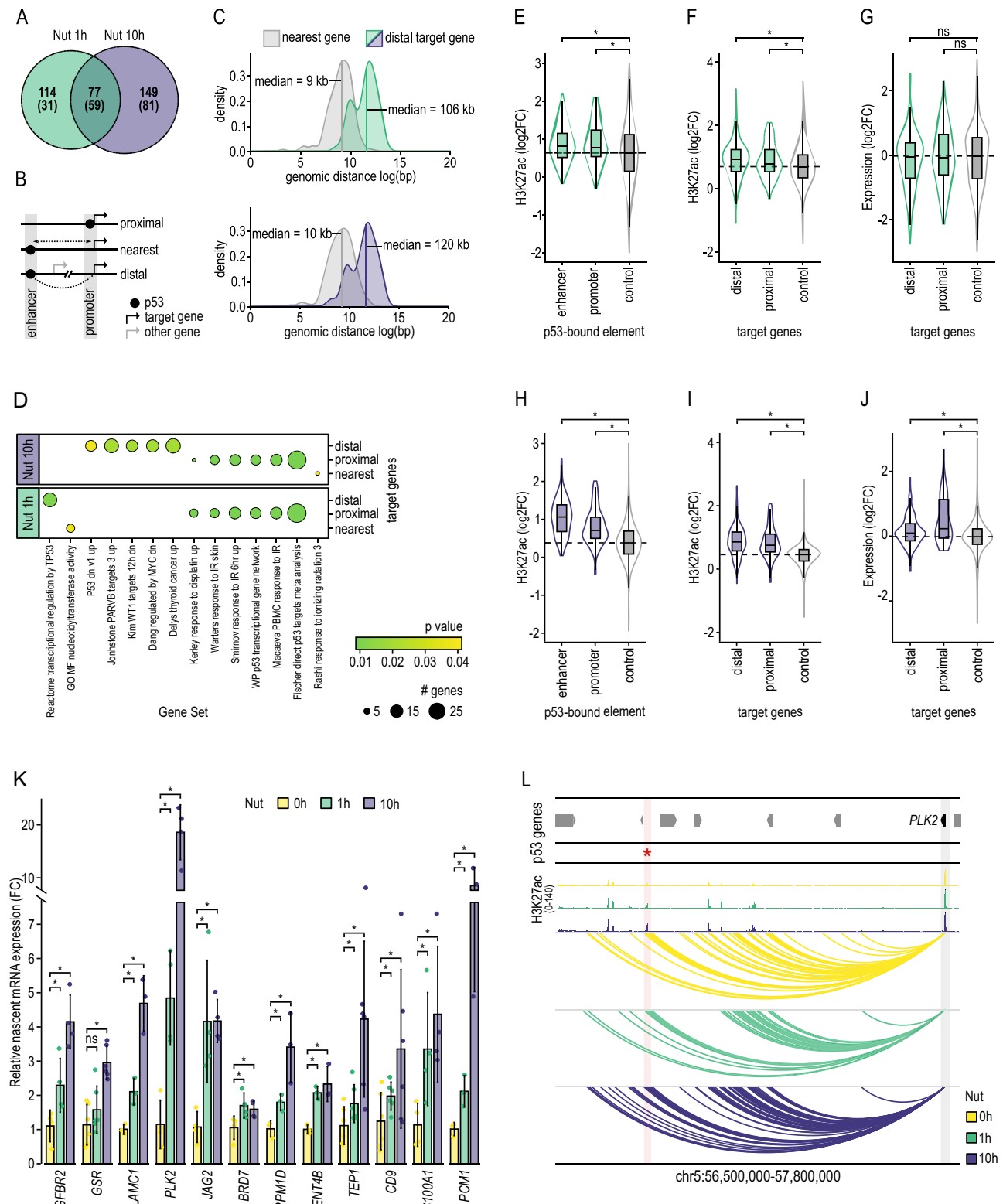

activation (Fig. 5E, H). Similarly, promoters of p53 distal target genes (i.e., gene promoters engaged in a DNA loop with a p53-bound enhancer) gained significantly more H3K27ac than genes that were not directly targeted by p53 (Fig. 5F, I). Moreover, these p53 distal target genes were upregulated at both the nascent transcribed (p value: 1e-8 after 1 h, and 1e-7 after 10 h, linear mixed model; see methods) (Suppl. Data 2) and mature messenger RNA levels (Fig. 5G, J, K). The *PLK2* kinase[37] gene is an example of a p53 distal target gene, which is controlled by a p53-bound

enhancer located 966 kb downstream from its transcriptional start site (Fig. 5L). *PLK2's* promoter and the p53-bound enhancer, both in spatial proximity facilitated by DNA looping, gained activity according to H3K27ac levels upon p53 activation. Besides, *PLK2* was significantly upregulated (Fig. 5K–L). However, in this same example, genes closer to the p53-bound enhancer did not respond to p53 activation. Together, these results provide evidence of p53's ability to drive transcription from a linear distal location.

**Fig. 5 | Identification of new p53 distal target genes through enhancer-promoter interactions. A** Overlap between p53 distal target genes at Nut 1 h and Nut 10 h time points. In parentheses are the number of p53 distal target genes which were primed (already interacting with a distal p53-bound enhancer prior to p53 activation) (i.e., Nut 0 h). Distal target genes are those genes engaging in PCHi-C interaction with a p53-bound enhancer. Nut 0 h refers to control cells without p53 activation. **B** Schematic representation of our definition of p53 target genes. "Proximal" target genes are those with p53 binding at their promoter (i.e., −1000/+200 nucleotides from the TSS). "Nearest" genes are those found closest in linear proximity to a p53-bound enhancer. "Distal" target genes are those found to interact with a distal p53-bound enhancer (defined using PCHi-C data). **C** Density distribution of the distance between the midpoint of a p53-bound enhancer and the transcription start site of the "nearest" gene (gray), or the transcription start site of the "distal" target gene linked by PCHi-C after 1 h (green) or 10 h (purple) of p53 activation. **D** Gene set enrichment analysis of gene sets from the Molecular Signatures Database (MSigDB) with significant enrichment (p-adj <0.05) in one or more p53 target gene groups. Bubble size indicates the number of genes found enriched for a given gene set, and green to yellow corresponds to the decreasing adjusted $p$ value for enrichment. $p$ values were calculated using a hypergeometric distribution test and adjusted for multiple comparisons using Benjamini-Hochberg (BH). **E** Differential H3K27ac between Nut 1 h and Nut 0 h time points (log2FC) at p53-bound regulatory elements (enhancer and promoter) and genome-wide (control). Genomic regions from p53-bound regulatory elements are excluded from genome-wide analysis. Two-sided Mann-Whitney test was used to test whether the mean H3K27ac log2FC differed significantly between groups using a $p$ value cutoff of 0.05 (star). Sample sizes are respectively 175, 82 and 476,760, for enhancer, promoter and control regions. **F** Differential H3K27ac between Nut 1 h and Nut 0 h time points (log2FC) at promoter elements of proximal and distal p53 target genes, and promoter elements of non-targeted genes (control). Two-sided Mann-Whitney test was used to test whether the mean H3K27ac log2FC differed significantly between groups using a $p$ value cutoff of 0.05 (star). **G** Differential expression between Nut 1 h and Nut 0 h time points (log2FC) of proximal and distal p53 target genes and non-targeted genes (control). Two-sided Mann-Whitney test was used to test whether the mean H3K27ac log2FC differed significantly between groups using a $p$ value cutoff of 0.05 (star). **F**–**G** Sample sizes are respectively 191, 82 and 14,233, for distal, proximal and control genes. **H, I** and **J**. As in (**E**, **F** and **G**), respectively, but considering target genes controlled by p53 at 10 h of activation. **H** Sample sizes are respectively 184, 77 and, 189,993, for enhancer, promoter and control regions. **I, J** Sample sizes are respectively 226, 77 and 13,803, for distal, proximal and control genes. **E**–**J** The boxes show the interquartile range (IQR), the central line represents the median, the whiskers add 1.5 times the IQR to the 75 percentile (box upper limit) and subtract 1.5 times the IQR from the 25 percentile (box lower limit). Significance calculated with two-sided Mann–Whitney test. The star indicates $p$ values below 0.05. **K** Relative nascent mRNA expression (fold change) of p53 distal target genes along p53 activation. A one-tailed Student's $t$ test was used to test whether relative expression differed significantly ($p$ value  < 0.05) between adjacent time points for each gene (star). A linear mixed model was used to test whether relative expression differed significantly between adjacent time points, globally. For this, relative expression was averaged between the three replicates at each time point for each gene ($p$ value: 6.5e-14 after 1 h, and 4.3e-13 after 10 h. linear mixed model see methods). Error bars correspond to one standard deviation of the mean. **L** Interaction landscape of *PLK2's* gene promoter (blue shade) showing stable interactions (arcs) with a p53-bound enhancer (red asterisk and pink shade) located 966 kb downstream. H3K27ac profiles along p53 activation are represented at the top. Peak tips colored in black represent a peak going over the scale limit. Arrows symbolize gene placement and orientation along the genomic window.

Finally, we investigate the cell type-specificity of the core transcriptional program of genes directly regulated by p53-bound enhancer over distance. To do so, we treated five different cancer cell lines wild type for p53 (Suppl. Fig. 6K) and three human primary cell types (Suppl. Fig. 7A) with 10 μM Nutlin-3a. Cells were collected at 0, 10, and 10 h of treatment and used to quantify the nascent transcript levels of well-known p53 target genes (i.e., *CDKN1A*, *TGFA*, *BAX*) and new target genes distally regulated by p53-bound enhancers that we previously identified (Suppl. Data 2). Globally, the p53-driven transcriptional response was conserved across cell types, whereas the kinetics and strength of the response vary between cell types and genes (Suppl. Figs. 6, 7), supporting the reliability of a new set of p53 target genes previously identified.

Altogether, our analysis demonstrates that DNA looping enables the effect of p53 activation to be broadcasted from enhancers to gene promoters and allows the identification of new p53 distal target genes, obtaining a more complete picture of the p53 gene regulatory mechanisms.

## p53 binding at enhancers leads to neo-loop formation with distal gene promoters

As we previously demonstrated, the promoter interactome was highly rewired during p53 activation (Fig. 4D). Next, we studied whether this rewiring could be caused, at least partially, by dynamic p53-mediated enhancer-promoter interactions. Only 47.1% (90/191) of p53 distal target genes at 1 h of activation were primed (i.e., gene promoter interacting with p53-bound enhancers prior activation) (Fig. 5A). This result suggests a novel role of p53 related to the establishment of DNA looping between enhancers and target gene promoters. Besides, just 23.6% (77/340) of distal target genes were in physical proximity with p53-bound enhancers at both time points (1 h and 10 h of Nutlin-3a treatment), most of which were primed (76.6% (59/77)).

Collectively, these results suggest the co-existence of two mechanisms by which p53 controls transcription of distal genes: i) a stable and primed mechanism that relies on a non-dynamic 3D chromatin structure, which could be associated with a core and stable transcriptional program; and ii) a dynamic and non-primed

mechanism associated with the formation and destruction of DNA loops along p53 activation, which could enable tailoring of the response to different types of stress or DNA damage.

We then focused the study of the p53-mediated promoter-enhancer interactome rewiring at early p53 activation (i.e., 1 h of Nutlin-3a treatment) where p53-indirect contributions (i.e., those related with factors transcriptionally controlled by p53) are minimal. Contrary to the general trend of loss of interactions (Suppl. Fig. 8A), p53-bound enhancers increased connectivity with promoters located a median distance of 100 kb away (Fig. 6A and Suppl. Fig. 8A–B). Specifically, 41.2% (91/221) p53-bound enhancers acquired interactions engaging distal target genes after 1 h of Nutlin-3a treatment, none of these being engaged with other genes before p53 activation (Fig. 6B). On the other hand, 25.8% (57/221) of p53-bound enhancers lost connectivity with any gene promoters.

Pathway-wise, genes that acquired de novo interactions with p53-bound enhancers within 1 h of p53 activation were enriched in the Reactome transcriptional regulation by *TP53* pathway[41] (e.g., *ATRIP*, *CARM1*, *DDIT4*, *GTF2H4*, *GSR*, *PRDX1*, *TCEB3*, *TP53INP1*, *YWHAH*) (Fig. 6C and Suppl. Data 9, 10). Besides, we also identified a number of target genes not previously associated with p53, which are distally controlled by p53 through new loop (neo-loop) formation and are not observed in control cells before p53 activation (e.g., *E2F2*, *DUSP4*, *SMAD1*, *DNAJA3*, *PPM1D*, *YTHDC2*, *CHD1L*, *KMT2E*, *KIAA1429*). For example, *JAG2* gene, regulator of the Notch signaling pathway, is rapidly upregulated (i.e., its promoter gains H3K27ac and its transcription is increased) by p53 via de novo formation of two DNA loops that engage one p53-bound enhancer each, located 186 kb and 185 kb away from *JAG2's* transcriptional start site at 1 h of treatment (Figs. 5K and 6D). This transactivation is transient since, at 10 h of p53 activation, both loops are erased, its promoter lost H3K27ac and *JAG2* reduced its transcription rate.

This highly dynamic spatio-temporal promoter interactome contrasts with a topologically stable regulation of both well-known and novel genes by p53 (e.g., *CDKN2A*, *PLK2*, *GTF2F2*). For instance, the primed spatial proximity through stable DNA looping of the *BRD7* promoter and the p53-bound enhancer correlated with a gain

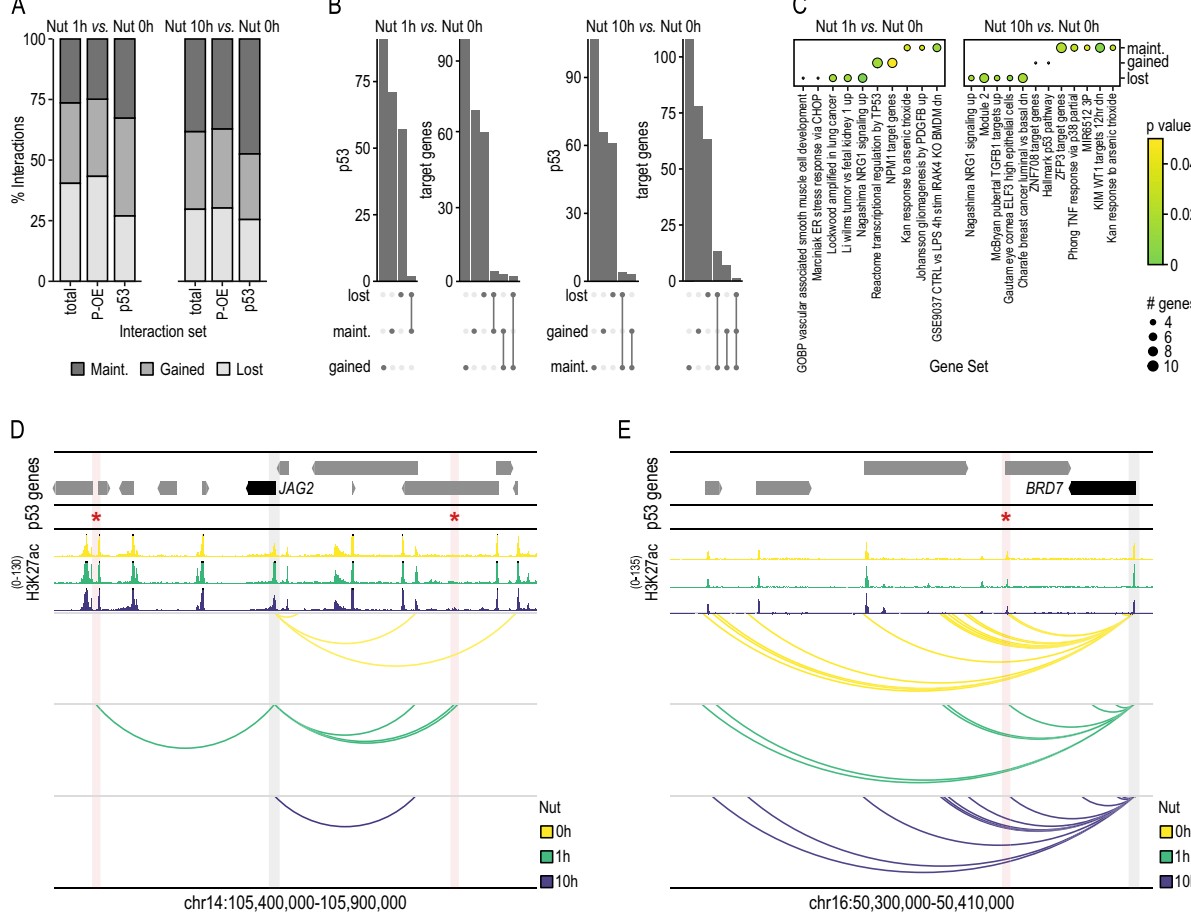

**Fig. 6 | p53 drives the formation of promoter-enhancer interactions or uses pre-established ones to control gene transcription over distance. A** Percentage of significant promoter interactions (CHiCAGO score ≥ 5) gained, lost, and maintained between Nut 1 h and Nut 0 h time points (left) and Nut 10 h, and Nut 0 h time points (right) respectively. The three interaction sets shown are comparing the total interactomes (total); comparing promoter interactions with non-promoter regions (promoter with other-end or P-OE); and comparing interactomes with a p53 binding site present at either end (p53). Nut 0 h refers to control cells prior to p53 activation. **B** Upset plot showing the overlap between p53 binding sites and p53 distal target genes found at maintained, gained and lost interactions between Nut 1 h and Nut 0 h time points (left) and Nut 10 h and Nut 0 h time points (right). **C** Gene set enrichment analysis of gene sets from the Molecular Signatures Database (MSigDB) with significant enrichment (*p*-adj <0.05) in one or more p53 distal target gene groups defined by the interaction dynamism between Nut 1 h and Nut 0 h (left) and

Nut 10 h and Nut 0 h time points (right). Bubble size indicates the number of genes found enriched for a given gene set. *p* values were calculated using a hypergeometric distribution test (or one-sided Fisher exact test) and adjusted for multiple comparisons using Benjamini-Hochberg (BH). **D** Interaction landscape of *JAG2's* gene promoter (blue shade) showing dynamic interactions (arcs) with two p53-bound enhancers (red asterisk and pink shade) located 186 kb and 185 kb away. H3K27ac profiles along p53 activation are represented at the top. Peak tips colored in black represent a peak going over the scale limit. Arrows symbolize gene placement and orientation along the genomic window. **E** Interaction landscape of *BRD7's* gene promoter (blue shade) showing stable interactions (arcs) with a p53-bound enhancer (red asterisk and pink shade). H3K27ac profiles along p53 activation are represented at the top. Peak tips colored in black represent a peak going over the scale limit. Arrows symbolize gene placement and orientation along the genomic window.

in activity according to H3K27ac and an increase in nascent transcription levels during p53 activation (Suppl. Data 2, 9 and Figs. 5K and 6E).

Collectively, these findings uncover an unexpected dependence of p53 on both newly formed and pre-existing enhancer-promoter loops to distally control gene transcription and ultimately determine the cellular response to cellular stress.

## Cohesin is required for the p53-mediated transcriptional response

As previously demonstrated, DNA looping enables long-distance broadcasting of p53 transactivation from enhancers to gene promoters. On the other hand, previous studies have shown that cohesin plays a pivotal role in organizing spatio-temporal chromatin architecture by stabilizing DNA loops[43]. Given these premises, we next explored the dependency between p53 on cohesin to trigger a transcriptional response. To do so, we took advantage of HCT116 cells in

which all alleles of the cohesin subunit *RAD21* were tagged with an auxin-inducible degron version 2 (AID)[44]. After validating RAD21's rapid degradation after 6 h of treatment with 1 μM of 5-Ph-IAA, we activated p53 using 10 μM of Nutlin-3a (Suppl. Fig. 9A, B). Treated cells were used to: i) confirm that DNA loops were erased using PCHi-C (Suppl. Fig. 9C, D and Suppl. Data 8); ii) generate high-quality ChIP-seq libraries to genome-wide profile the H3K27ac distribution, which allowed us to characterize active promoters (Suppl. Fig. 7E, F and Suppl. Data 4); and iii) analyze nascent transcript by Real-Time Quantitative Reverse Transcription Polymerase chain reaction (qRT-PCR), which allowed the measurement of gene transcriptional activity (Suppl. Data 2).

Focused on the previously identified 340 p53 distal target genes (Fig. 5A, B and Suppl. Data 9), most of the DNA loops engaging p53-bound enhancers with these distal target gene promoters were erased (67%) or found to reduce their interaction strength (76%) upon cohesin depletion (Fig. 7A, B, G, H), being the shorter-range DNA loops those

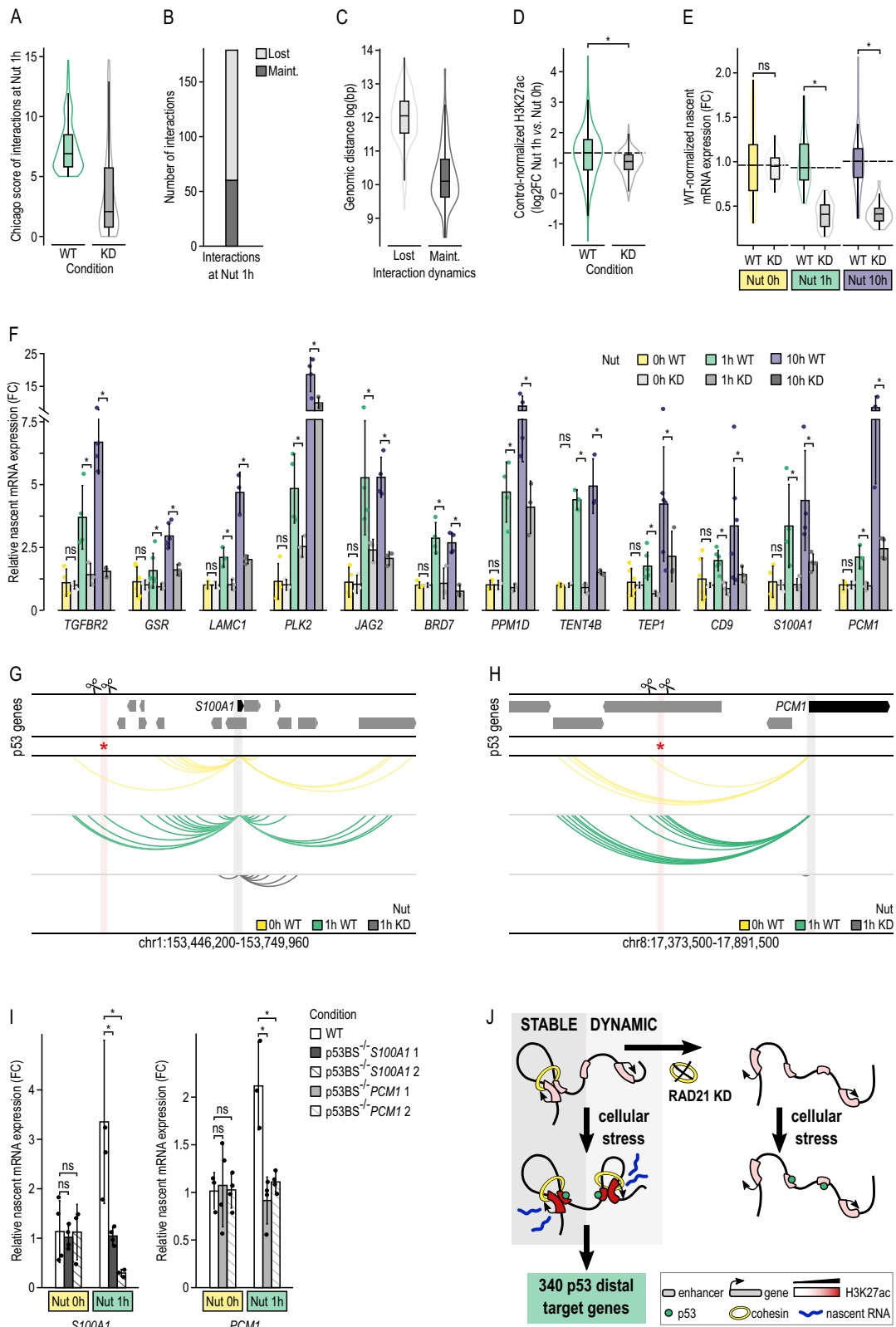

identified as less dependent on RAD21 depletion, as previously reported[22] (Fig. 7C, G, H). We observed that RAD21 degradation hindered H3K27ac gain at their promoters during early p53 activation compared with control genes, as opposed to the significant gain of H3K27ac observed in wild-type conditions (Fig. 7D). In addition, we observed a significant drop in transcription rates depicted by the relative nascent mRNA expression levels (*p* value: 0.6 at Nut 0 h, 1.4e-13

at Nut 1 h and 2.0e-15 at Nut 10 h) (Fig. 7E, F). For instance, RAD21 degradation impeded p53-mediated transcriptional upregulation of tumor suppressor gene *PLK2*[37] (Fig. 7F). Similarly, transcriptional upregulation of *LAMC1*, which encodes for a Laminin Subunit involved in cell adhesion, differentiation and metastasis[45], as well as *TENT4B*, which encodes for a nucleotidyltransferase involved on RNA metabolic processes[46], were also compromised (Fig. 7F).

**Fig. 7 | Cohesin depletion impedes p53-mediates transcriptional response.**
**A** Distribution of CHiCAGO scores of the 179 interactions found between p53-bound enhancers and distal target genes found at Nut 1 h for wild-type (WT) and RAD21 depleted (KD) conditions. **B** Number of significant promoter interactions (CHiCAGO score ≥ 5) lost and maintained between WT and KD conditions at Nut 1 h. **C** Logarithmic distribution of genomic distance of interactions lost ($n = 135$) and maintained ($n = 68$) at Nut 1 h between WT and KD conditions. **D** Control-normalized differential H3K27ac (log2FC) at promoters of p53 distal target genes between Nut 1 h and Nut 0 h time points in wild type (WT) and RAD21 depleted (KD) conditions. Control genes are defined as those not directly targeted by p53. Two-sided Mann-Whitney test was used to test whether the control-normalized mean H3K27ac log2FC differed significantly between groups using a $p$ value cutoff of 0.05 (star, $p$ value = 3.2e-6). Sample sizes are 191 promoters for WT and 184 promoters for KD. **E** Distribution of wild-type-normalized nascent mRNA expression levels (fold change) of 12 distal target genes along p53 activation in WT or KD conditions, from at least three replicates each. A two-sided Mann-Whitney test was used to test whether expression levels differed significantly between conditions for each time point (star) ($p$ value: 0.6 at Nut 0 h, 1.4e-13 at Nut 1 h and 2.0e-15 at Nut 10 h). **A**, **C**–**E** The boxes show the interquartile range (IQR), the central line represents the median, the whiskers add 1.5 times the IQR to the 75 percentile (box upper limit) and subtract 1.5 times the IQR from the 25 percentile (box lower limit). Significance calculated with two-sided Mann–Whitney test. The star indicates $p$ values below 0.05. **F** Relative nascent mRNA expression (fold change) of p53 distal target genes

along p53 activation in the presence (WT) or absence of RAD21 (KD). A one-tailed Student's $t$ test was used to test whether relative expression differed significantly between conditions at each time point for each gene (star). A linear mixed model was used to test whether relative expression differed significantly between conditions of adjacent time points, globally. For this, relative expression was averaged between the three replicates at each condition and time point for each gene ($p$ value: 0.3 at Nut 0 h, and 2.3e-21 at Nut 1 h and 4.3e-16 at Nut 10 h. linear mixed model see Methods). Error bars correspond to one standard deviation of the mean. **G** Interaction landscape of *S100A1*'s gene promoter (blue shade) showing interactions (arcs) with a p53-bound enhancer (red asterisk and pink shade) located 101 kb upstream. Arrows symbolize gene placement and orientation along the genomic window. **H** Interaction landscape of *PCM1*'s gene promoter (blue shade) showing interactions (arcs) with a p53-bound enhancer (red asterisk and pink shade) located 202 kb upstream. Arrows symbolize gene placement and orientation along the genomic window. **I** Levels of relative nascent mRNA expression at 0 and 1 h after p53 activation, measured by qRT-PCR, for *S100A1* (left), in wild type cell and both clones of CRISPR deletions targeting the p53 binding site distally acting on *S100A1* (p53BS−/− *S100A1*); and for *PCM1* (right), in wild type cells and both clones of CRISPR deletions targeting the p,53 binding site distally acting on *PCM1* (p53BS−/− *PCM1*). **J** Graphical representation of both mechanisms by which p53 controls transcription of distal genes: i) a stable and primed mechanism that relies on a non-dynamic 3D chromatin structure; and ii) a dynamic and non-primed mechanism associated with the formation and destruction of DNA loops along p53 activation.

Summarizing, these results disclose a novel association between p53 and cohesin in the context of cellular stress and demonstrate that RAD21 degradation reduces p53-mediated distal promoter activation and compromises p53-driven transcription response.

## CRISPR-Cas9-based functional validation of new p53 direct target genes

Finally, we formally proved the new mechanistic model that we propose being that p53 controls transcription of distal target genes via DNA looping engaging p53-bound enhancers and gene promoters. To do so, we selected two target genes (i.e., *S100A1* and *PCM1*) controlled by a single p53-bound enhancer each, located 101 kb and 202 kb away from their promoter region, respectively (Fig. 7G, H). Using CRIPR-Cas9 methodology, we generated homozygotic clones lacking the p53 sequence-specific DNA motif (Suppl. Fig. 9G), which were treated with Nutlin-3a. After validation of p53 activation by qRT-PCR and WB (Suppl Fig. H, I), we quantified nascent transcription. As shown in Fig. 7I, both distal target genes were found to no longer transcriptionally respond to p53 activation after deletion of the distal p53 sequence-specific DNA motif identified as the regions bound by their targeting p53.

Collectively, these results support reliability of the set of new p53 direct target genes distally controlled by p53-bound enhancers and uphold the mechanistic model by which DNA looping enables the effect of p53 activation to be broadcasted from enhancers to gene promoters to ultimately trigger a transcriptional response (Fig. 7J).

## Discussion

Although the tumor suppressor p53 was first characterized as a transcription factor 44 years ago[47–57], there are still many unresolved questions about its mechanism of action, including its interplay with the 3D genome organization to ultimately trigger the activation of hundreds of genes in a highly coordinated fashion. Here, using a fine-scale multi-omics time course, we demonstrated that p53 activation drives dramatic genome-wide changes in genome compartments, TADs and DNA loops to ultimately trigger a temporally dependent transcriptional response against cellular stress.

Unexpectedly, we observed that the most changes in spatio-temporal genome organization occur at 1 h of p53 activation, prior to any dramatic p53 protein activation, significant cell cycle alterations or apoptosis. This finding agrees with a previous study that identified ~200 genes being rapidly activated only 30 min after p53 activation using Global Run-on Sequencing (GRO-Seq) analysis[58]. However, with

few exceptions[59–61], most genome-wide alterations in 3D genome architecture associated with any cellular perturbation have been reported at later time points, suggesting a slower chromatin dynamism or advising the redesign of these time-resolved studies to capture early events. These changes in spatio-temporal genome organization include direct p53 effects, immediately associated with its binding to DNA, and indirect p53 effects driven by factors transcriptionally regulated by p53. For instance, we have identified chromatin remodelers (e.g., *BRD7*, *CHD1L*), transcriptional co-activators (e.g., *CARM1*), transcription factors (e.g., *E2F2*, *SMAD1*), components and regulators of the transcriptional machinery (e.g., *GTF2H4*, *TCEB3*, *GTF2F2*) and epigenetic modifiers (e.g., *KMT2E*) as p53 distal target genes that may also contribute to DNA topological alterations. Direct p53 effects, which mainly contribute to dynamism at 1 h of p53 activation, are primarily associated with gain of compartmental activity and enhancer-promoter loops. Here, we emphasize that these effects occur in early response, when the confounding effects of indirect transcriptional and post-transcriptional regulation are minimal. Our findings support the role of p53 as a transcriptional activator and demonstrate a previously unappreciated architectural role as a regulator at distinct topological layers. Contrarily, indirect p53 effects could mainly contribute to later dynamism and are more heterogenous depending on the activating or repressing nature of the downstream effectors.

The identification of reliable and reproducible p53 direct targets genes has been historically challenging due to the misleading association of p53-bound enhancers to target genes based on sequence proximity. Indeed, previous studies only considered as potential p53 direct targets those genes found differentially expressed following p53 activation or inactivation, whose transcriptional start sites are located from 5[62] to 100 kb[63] away from a p53 binding event. However, enhancers have the capacity to regulate genes located up to a few megabases away[64–66] and p53-bound enhancers should not be an exception to the rule. To overcome this limitation, in this work, we made use of PCHi-C[67], a methodological breakthrough that we developed to associate distal regulatory elements and target genes in a genome-wide, unbiased manner[68,69]. Specifically, we identified 340 p53 distal target genes, including previously defined target genes as well as new ones. In fact, using as reference a survey of 346 target genes derived from 319 individual gene studies[40] and a set of 3509 target genes derived from 16 high-throughput data sets[40], only 13 and 89 of our p53 distal target genes, respectively, were previously identified, in part due to the bias

of analyzing only proximal regions to p53-binding sites and promoters. Indeed, p53 distal target genes that we identified were located at a median distance of 116 kb from the functional p53 binding sites, outside of the largest arbitrary window of 100 kb previously established. Moreover, only 7% of the p53 distal target genes that we identified were found to be the nearest gene of the targeting p53-bound enhancers. These results, all together, showcase the power of our long-range promoter interaction data to provide a more comprehensive landscape of the p53 direct transcriptional regulome. Another limitation in the identification of the true set of p53 target genes is the frequent use of late time points of p53 activation. This strategy unavoidably includes confounding indirect effects and does not capture early p53 targets genes dynamically regulated in time. Finally, identification p53 direct targets has historically been performed using poly-A mature RNA-seq (mRNA) analysis along p53 activation. However, as we demonstrated, mRNA levels do not fully capture the p53-driven transcriptional effects, hindering the identification of truly direct p53 targets. We think that the discrepancies between steady state levels of mature RNAs and nascent transcripts levels that we mainly identified at the early time point could be explained by the p53-driven transcriptional upregulation of micro RNAs and RNA binding proteins (RBPs) that could ultimately affect mRNA stability and steady-state levels of mature RNAs. Specifically, 7 microRNAs and 84 RBPs are significantly upregulated upon 1 h of p53 activation. Among these, two microRNAs (miR-1293 and miR-1915) and nine RBPs (*SRSF1, EIF3, BRBM4, RBM14, ESRP2, RBM12B, PUF60, DNAJC17*, and *EIF3B*) are distally controlled by p53-bound enhancers. Moreover, miR-4738 has a functional p53 binding at its promoter region.

In this study, we have also shown that the p53 regulome is characterized by extensive rewiring of the p53-bound enhancer-promoter loops. Previous studies[6–8,70] proposed a mechanism, also called the permissive model, by which p53 relies on pre-programmed and stable enhancer–promoter loops to rapidly respond to p53-activating stimuli[71]. However, these studies used chromosome conformation capture technology to analyze only a few p53-bound enhancer-promoter loops. Using an unbiased genome-wide high-resolution approach, we have shown the co-existence of an additional dynamic and non-primed mechanism, called the instructive model, by which p53 controls transcription of distal genes through the formation and destruction of p53-bound enhancer-promoter loops (Fig. 7J). The permissive model could be associated with a rapid core transcriptional response structurally stable in time. Contrarily, the instructive model could be associated with a dynamic transcriptional response over time and may enable tailoring the response to different types of stress.

p53 binds a core set of regions regardless of cellular and epigenetic contexts via high-affinity recognition of its binding motif without the need of auxiliary transcription factors[27]. However, despite its largely invariant binding, p53 leads a highly cell-type-specific and stimulus-specific transcriptional response previously associated with permissiveness of the epigenetic environment[28,29]. On top of the dynamism along p53 activation, p53-bound enhancer-promoter loops could also be cell-type and stimulus-specific, restricting the p53 targets available for direct transactivation. This type of structural guidance would ultimately allow tailoring the response to different types of stress responses and pathway activations. Further investigations to decode the cell-type-specific and stimulus-specific variation of the p53-centric spatio-temporal genome architecture will be needed to fully understand the variable p53 transcriptional response.

Emphasizing the relevance of spatio-temporal genome architecture, here we showed that cohesin was critically required for the induction of distal target gene expression triggered by p53 activation. Since cohesin removal leads to deletion of promoter-enhancer interactions[10,14], our findings reinforce the critical role of DNA loops in mediating transcriptional changes in response to cellular stress. Paradoxically, cohesin depletion has little impact on steady-state gene

transcription and enhancer activity[10]. However, two recent studies have implicated cohesin in inducible gene transcription. Specifically, it has been demonstrated that RAD21 removal hinders macrophages' transcriptional response to the endotoxin lipopolysaccharide stimulation[23] and prevents neurons from establishing transcriptional programs in response to new stimuli[24]. In agreement with these results, our findings demonstrate that cohesin is critical to support changes in transcriptional response, as opposed to the maintenance of a stable transcriptional landscape (Fig. 7J).

Although in this study we have been focused on the cohesin-dependent loop extrusion model, it is important to highlight that p53 has an intrinsically disordered domain[72]. Intrinsically disordered proteins have the capacity to undergo liquid-liquid phase separation to form membrane-less droplets that act as transcriptional condensates, promote gene expression, and remodel 3D chromatin organization[73]. However, to what extent p53 has the ability to induce topological changes in genome architecture through this mechanism and what the transcriptional consequences would be remains unknown.

Finally, our findings not only advance our understanding of p53 biology, but also pave the way for better-informed p53-based cancer therapies. Tumor suppressor gene *TP53* is the most frequently mutated gene in cancer[74]. p53-based therapy can either re-establish the functionality of mutant p53 proteins or safeguard wild-type p53 from degradation in the cases of over-activity of its negative regulators (e.g., HDM2, HDM4). However, only a few of these therapeutic strategies have reached late-stage clinical trials for various types of cancer[75–77] and no drugs have been approved in Europe or USA up to date[78]. One of the multiple reasons behind the limited success of this strategy to safeguard wild-type p53 could be the presence of loss-of-function mutations affecting p53 distal target genes. In this scenario, treatments to block p53 degradation would not be effective since the p53-mediated transcriptional response is compromised in a p53-independent manner. For this reason, a comprehensive list of true p53 direct target genes may help to better predict which patient will respond based on preservation of the wild-type cascade downstream of p53. Another potential reason of failure of p53-based therapeutic approaches could be associated with mutations affecting cohesin. Although cohesin mutations are frequent in several types of cancer[74,79,80], the mechanisms by which these mutations trigger cancer development and progression are not completely understood[81]. The Cancer Genome Atlas project has reported a comprehensive landscape of molecular alterations in bladder cancer, which includes mutations affecting p53 and cohesin in 49% and 11% of the cases, respectively[82]. Since our study disclose a p53 dependency on cohesin to drive a tumor suppressor transcriptional response, further investigations of mutually exclusive or synergistic effects of p53 and cohesin mutations are required to ultimately guide better therapeutic decisions.

Altogether, our study implicates spatio-temporal genome architecture as an instructive force for implementing a p53 transcriptional response to cellular stress, identifies a new set of reliable p53 direct target genes, and may help the future design of better-informed p53-based cancer therapies.

## Methods
### Cell culture
HCT116 (RRID:CVCL_0291), HCT116-RAD21-mAC (kindly provided by Dr. Masato T. Kanemaki), SH-SY5Y (RRID: CVCL_0019), and HepG2 (RRID:CVCL_0027) cell lines were cultured in RPMI1640 medium with Glutamax (Fisher Scientific #61870044) supplemented with 10% FBS (Life Technologies #10270106) and penicillin-streptomycin (Dutscher #L0022-100). MCF-7 (RRID:CVCL_0031), C32 (RRID:CVCL_1097), and CAKI-1 (RRID:CVCL_0234) were cultured in DMEM-Glutamax (Life-technologies cat#61965-026) supplemented with 10% FBS and penicillin-streptomycin. Human Umbilical Vein Endothelial Cells (HUVEC) (Lonza, C2519A) and Human Dermal Lymphatic Endothelial

Cells (HDLEC) (Promocell, C-12217) were cultured in 0.5% gelatine-coated culture well in EGM2 medium (PromoCell, C-22211) supplemented with Endothelial Cell Growth Medium 2 KIT (Promocell, C-22111), 10% FBS (Gibco, 10270-106), 1% penicillin/streptomycin (Gibco, 15140-122). Human Pericytes from Placenta (hPC-PL) (Promocell, C-12980) were cultured in Perycte growth media (Promocell, C-28041) with supplement mix, 10%FBS (Gibco, 10270-106), 1% penicillin/streptomycin (Gibco, 15140-122).

Cell cultures were kept in a humid incubator at 37 °C at 5% CO2. Besides, to activate p53 response, cells were supplemented with 10 μM of Nutlin-3a (Quimigen #HY-10029) dissolved in DMSO and incubated for the required time. To reverse p53 response, Nutlin-3a treated cells were washed twice with PBS1x, and cultured in completely fresh growth media for an additional 24 h. Degradation of the AID2-tagged RAD21 protein was induced by addition of 1 μM of 5-Ph-IAA (MedChemExpress #HY-134653) dissolved in DMSO for at least 6 h prior to further experimental procedure. Appropriate negative controls for each condition were established by incubating the cells with the drug vehicle (DMSO).

## p53 binding site CRISPR-targeting
Single guide RNA (sgRNA) sequences targeting p53 binding sites were obtained from the GRCh37.p13 genome. Two sgRNAs surrounding the p53 binding motif (length 222 bp and 224 bp for the binding sites associated with *S100A*1 and *PCM1*, respectively) and with high predicted cleavage (On-target score >60[83] and Off-target score >50[84] were selected using the "CRISPR Guide RNA Design Tool" (https://www.benchling.com). For the different loci of interest to be deleted, sgRNAs for each side were cloned into pX330-GFP-hSpCas9 (https://www.addgene.org/128385/) and pX330-BFP-hSpCas9 (https://www.addgene.org/64323/) respectively. Precise cloning was validated by Sanger sequencing before gene targeting experiments. Sequences of the oligonucleotides used are provided in Suppl. Data 11.

According to the manufacturer's instructions, HCT116 cells were transfected by nucleofection using Kit V and program D-032 (Amaxa Nucleofector, Lonza). A mix of 1 μg of each plasmid (pX330-GFP and pX330-BFP) in 100 μl of nucleofection solution was used to transfect $1 \times 10^6$ cells. Two sets of nucleofection were done for each condition and cells were pooled together for recovery. The day after, cells were washed with PBS1x to remove cellular debris, and GFP/BFP double-positive cells were selected among the population by single-cell FACS sorting. After expanding the GFP+/BFP+ clones, their DNA was extracted (Wizard Genomic DNA Purification Kit (Promega #A1120)) and analyzed by PCR to detect clones carrying bi-allelic deletions on the regions of interest.

## Cell cycle analysis by FACs
After incubation with drugs or vehicle, $10^6$ cells per condition were collected in a single cell suspension, washed in PBS1x, fixed by adding 0.9 ml of ice-cold 70% ethanol dropwise while vortexing and stored at −20 °C for at least 24 h. On the day of the analysis, cells were spun down at 1000 g, 5 min and washed with PBS1x. Cells were stained for 30 min at room temperature using FxCycle PI/RNase Staining Solution (Life Technologies #F10797) following manufacturer's instructions. Cell cycle was assessed using BD FACSCanto™ II flow cytometer and data was analyzed using the Watson Pragmatic algorithm platform provided by FlowJo v10[85].

## Western blot
For validation of drug treatments, at least $5 \times 10^6$ cells from different conditions were harvested, and washed with cold PBS1x. Cytoplasmatic membranes were lysed with 0.1% Triton X-100 (AppliChem #A4975,0100) diluted in ice-cold Sucrose Buffer (0.32 M Sucrose, 10 mM Tris-HCl pH8, 3 mM CaCl2, 2 mM Magnesium Acetate, 0.1 mM EDTA) supplemented with Pierce™ Protease and Phosphatase

Inhibitor Mini Tablets, EDTA-free (Thermo Scientific #A32961). Cytosolic fractions were collected after centrifugation at $1000 \times g$, 3 min, 4 °C; and nuclei were lysed 30 min at 4 °C in rotation, in ice-cold RIPA buffer (50 mM TisHCl pH8, 150 mM NaCl, 1%NP40, 0.5% Na-deoxycholate, 0.1%SDS) containing Pierce™ Protease and Phosphatase Inhibitor Mini Tablets, EDTA-free (Thermo Scientific #A32961) and 1 mM DTT (Merck Life Science #DO632- 1G). Cell lysates were sonicated using a UP50H Hielscher sonicator (1cy, 90%amp, 10 s per burst, 3 bursts per sample) and clear cell extracts were collected after centrifugation at max speed, 8 min, 4 °C. Total cell lysates were quantified using BCA Pierce™ (Thermo Scientific #23225) and at least 20 μg of protein were resolved on 12 or 15% polyacrylamide gels, transferred onto nitrocellulose membranes and incubated with appropriate primary and secondary antibodies. The following primary antibodies were used: p53 DO1 (Santa Cruz Biotechnology, sc-126, diluted 1:200), p53 (phospho S46) (Abcam, ab76242, diluted 1:5000), CDKN1A (Abcam, ab109520, diluted 1:1000), RAD21 (Abcam, ab992, diuted 1:1000), Vinculin (Abcam, ab129002, diluted 1:10,000), H3 (Thermo Fisher Scientific, PA5-16183, diluted 1:4000), α-Tubulin (SIGMA-ALDRICH, T6199, diluted 1:10,000). The following secondary antibodies were used: IRDye 800CW Goat anti-rabbit IgG secondary antibody (LICOR #926-32211, dilute 1:10,000) and IRDye 680RD Goat anti-mouse IgG secondary antibody (LICOR #926-68070, dilute 1:10,000).

Immunoblots were imaged using an Odyssey CLx imager and Image Studio Lite v5.2. For the quantification, optical density was measured using Image Studio Lite Version 5.2 and normalized by that of its housekeeping protein (Suppl. Data 1). Plots show fold change of each condition versus Nut 0 h (DMSO-treated) cells. Statistical significance was tested in fold change ratios using a non-parametric Mann-Whitney test ($p < 0.05$).

## In situ Hi-C and PCHi-C libraries generation
Hi-C and PCHi-C libraries were prepared as described in ref. 68. Briefly, cells were fixed in DMEM medium supplemented with 10% FBS and 2% methanol-free formaldehyde for 10 min rotating at room temperature. After quenching the formaldehyde with 0.125 M glycine and washing the cells with 1X PBS, nuclei were extracted by lysing the cells in 10 mM Tris-HCl pH 8.0, 10 mM NaCl, 0.2% IGEPAL CA630 (SIGMA-ALDRICH #18896-50 ML), 1× cOmplete EDTA-free protease inhibitor cocktail. Chromatin was digested overnight with HindIII enzyme and the cohesive restriction fragment ends were filled in with dCTT, dTTP, dGTP and biotin-14-dATP (Invitrogen #19524-016) nucleotides. After blunt-end ligating the restriction fragment ends, the chromatin was decrosslinked by incubating overnight with proteinase K and purified by phenol:chloroform:isoamyl alcohol 25:24:1 (SIGMA-ALDRICH #P3803-100ML) extraction.

Non-informative biotin at restriction fragment ends was removed by incubating the samples with dATP and T4 DNA polymerase (New England Biolabs #M0203S). After purifying the DNA again with phenol:chloroform:isoamyl alcohol extraction, 10 μg of the chromatin was sheared using a Covaris M220 focused ultrasonicator in 130 μl cuvettes using the following parameters: 20% duty factor, 50 peak incident power, 200 cycles per burst, 65 s. The ends were end-repaired and dATP-tailed, followed by a biotin-pulldown using Dynabeads MyOne streptavidin C1 paramagnetic beads (Thermo Fisher #65001) to enrich for those DNA fragments which contain information of a chromatin loop. PE Illumina adapters (Suppl. Data 11) were ligated to the sample and the library was amplified for eight cycles. Finally, the library was purified using a SPRI bead double-sided selection (0.4−1 volumes). The size and concentration of the finished Hi-C libraries were assessed by DNA ScreenTape Analysis (Agilent #5067-5582) on an Agilent 2200 Tapestation.

For PCHi-C libraries, 500−1000 ng of Hi-C library were captured using the SureSelectXT Target Enrichment System for the Illumina Platform (Agilent Technologies) as instructed by the manufacturer.

Captured library was amplified a total of 4 cycles, and the size and concentration of the finished PCHi-C libraries were assessed by high-sensitivity DNA ScreenTape Analysis (Agilent #5067-5584) on an Agilent 2200 Tapestation.

## ChIP-seq library preparation

Cells were crosslinked in 1X PBS supplemented with 1% methanol-free formaldehyde (Thermo Fisher #28908) for 10 min rotating at room temperature. After quenching the formaldehyde with 0.125 M glycine for 5 min rotating at room temperature. After washing the cells with 1X PBS, pelleted cells were lysed in 1% SDS (AppliChem #A0676,0250), 10 mM EDTA (Invitrogen #AM9260G), 50 mM Tris-Cl pH 8.1 (Invitrogen #AM9855G) at 4 °C for 20 min. Samples were sonicated using a Covaris M220 focused ultrasonicator at a concentration of $20 \times 10^6$ cells/ml using the following parameters: 10% duty factor, 75 peak incident power, 200 cycles per burst, 15 min. After centrifuging the samples at $18,407 \times g$ to remove cell debris and recovering the chromatin-containing supernatant, 33 µl of sonicated chromatin were prepared for immunoprecipitation by adding 267 µl of buffer containing 1% Triton (AppliChem #A4975,0100), 1.2 mM EDTA, 16.7 mM Tris pH 8, 167 mM NaCl (Invitrogen #AM9760G), 1X cOmplete protease inhibitor cocktail (Merck cat. #11873580001) and the appropriate amount of antibody (1ug of α-H3K27ac and 0.5 µg of α-H3K4me1; Diagenode #C15410196 and #C15410194, respectively). Samples were incubated rotating at 4 °C overnight. Chromatin was immunoprecipitated by adding 10 µl of both protein A and protein G-conjugated paramagnetic beads (Invitrogen #1001D and #1003D, respectively) and incubated at 4 °C for 1 h. After washing the beads, chromatin was decrosslinked overnight at 65 °C using proteinase K (ThermoFisher Scientific #EO0491) and purified using 1.1 volumes of SPRI beads according to the manufacturer's instructions (CleanNA #CNGS-0050). For ChIP inputs, an equivalent amount of sonicated chromatin was directly decrosslinked and purified as before.

ChIP-seq libraries were performed using the KAPA HyperPrep Kit (Roche #07962363001) according to the manufacturer's instructions using Truseq Illumina adapters and PCR primers described in Suppl. Data 11. Samples were sequenced to reach a minimum number of either 20 M or 45 M valid paired-read for H3K27ac and H3K4me1 histone marks respectively.

## RNA-seq library preparation

RNA was extracted from 200,000 frozen cell pellets using the RNAeasy Mini Kit (Qiagen #74104) following the manufacturer's instructions. Total RNA's integrity and concentration were assessed using RNA ScreenTape Analysis (Agilent #5067-5576) on an Agilent 2200 Tapestation. Samples were sequenced to reach a minimum number of 30 M unique valid paired reads.

## qRT-PCR

RNA was extracted from the corresponding conditions from 200,000–250,000 cell pellets using the RNeasy Mini Kit (Qiagen #74104) according to the manufacturer's instructions. On-column DNase (Qiagen #79254) treatment was applied to ensure no traces of genomic DNA were carried over during the genomic purification. The RNA was retrotranscribed using the SuperScript III First-Strand Synthesis System kit (ThermoFisher #18080051) according to the manufacturer's instructions using random hexamer priming and the final cDNA was further diluted to 3.7 ng/ul.

qRT-PCR primers for a given gene were designed against the most common intron between alternative transcripts (merged Ensembl/Havana database for protein coding transcripts) using Primer3 web tool (v.4.1.0 https://primer3.ut.ee/) using the following parameters: product size ranges 100–150, optimal primer Tm 60 °C, the rest in default. The primers were further checked for unique hybridization on the genome using the UCSC BLAT web tool using the GRCh37/hg19

human version of the genome, and tested for their amplification efficiency (between 85 and 115%, Suppl. Data 11).

The qRT-PCR reactions were carried out on 384-well plates using a QuantStudio 7 Flex Real-Time PCR System (ThermoFisher #4485701) using 10 µl reactions (2 µl of cDNA 3.7 ng/µl, 0.5 µl of forward and reverse primer mix 10 µM each, 5 µl of SYBR Green PCR Master Mix (ThermoFisher #4368577) and 2.5 µl of nuclease-free water). Raw data was analyzed using the QuantStudio software (v.1.3) and the nascent transcript expression fold change was calculated using the ΔΔCt method[86] against *HPRT1* as the housekeeping gene. Primer sequences are detailed in Suppl. Data 11, and nascent transcript expression fold change data are summarized in Suppl. Data 2.

We adjusted a linear mixed model to relative nascent mRNA expression to decide whether gene expression can be explained by the fixed-effect of the experimental condition (either the effect of Nutlin-3a over time or degradation of RAD21), or by a random effect. *p*-values correspond to the rejection of a hypothesis which states that the random effect fits better than the experimental condition.

## Hi-C processing

Various steps were taken to process the Hi-C data, including read quality control, mapping, interaction detection, filtering, and matrix normalization using the TADbit pipeline[25] (specific version: https://github.com/fransua/TADbit/tree/p53_javierre). The mapping of di-tags was performed using GEM3 mapper[87] onto the GRCh37.p13 reference genome (hg19 downloaded from UCSC, http://genome.ucsc.edu). TADbit mapping consists of two steps. First full reads are mapped, then, for the remaining unmapped reads, TADbit searches for HindIII ligation sites consisting of facing fragments of restriction-enzyme (RE) sites (e.g., AAGCTAGCTT). These reads with ligation sites are then split and their original HindIII site is reconstructed. Second, each of the read fragments undergoes a second round of mapping. This methodology may result in reads mapped in more than the two locations expected for Hi-C di-tags. TADbit crumbles these multiple mappings (see Suppl. Data 3) into multiple individual di-tags.

Di-tag filtering was then conducted on pairs of mapped read fragments in order to remove experimental or computational artifacts. Filters used in TADbit were 1, 2, 3, 4, 6, 7, 9, and 10 (see TADbit online documentation); namely we filtered for di-tags in the same RE-fragment (1: self-circle, 2: dangling-end, and 3: mapping-errors); or in contiguous fragments (4: extra dangling-end); for di-tags mapped in uninformative RE fragments, either too short, below 50 nucleotides, or too large, above 100 kb (6: too short, 7: too large); for PCR artifacts (9: duplicated); and finally, we also filtered for di-tags with at least an end mapped too far from any RE-site, as Hi-C product should necessary come from their vicinity (10: random breaks). The specific distance considered to fall in this last category is automatically defined by TADbit using the length distribution of single-fragment di-tags (specifically di-tags falling in the second filer dangling-end).

Hi-C interaction matrices at 50 kb and 100 kb resolutions were generated from filtered di-tags. These matrices are built, for each time point, by combining biological replicates. Interaction matrices were then cleaned by removing bins (rows or columns) with a ratio of mid-range cis interactions (interactions below 5 Mb) over total interactions below 1. Bins with more than expected long-range interactions were considered artefactual. The genomic interaction matrix was normalized using the ICE normalization as defined in ref. 88.

A/B compartments were called independently for each chromosome and time point at a 100 kb resolution. To this end, ICE-corrected interaction matrices were further distance corrected and finally transformed into Pearson correlation matrices[89]. In order to reduce potential noise in our data matrix we used a median filter with a size of 3 bins, similar to the methodology implemented in the HOMER software[90]. For each chromosome, we computed the three leading eigenvectors and performed a manual assessment to determine which

one potentially provided a more accurate representation of the heterochromatin and euchromatin segregation. The parameters considered for the decision of the eigenvector to use were: the correlation with GC-content (expected to be high in A-compartments), the enrichment of our available markers for activity (H3K27ac and RNA-seq), the relative importance (eigenvalue) of each of the eigenvectors, the correlation between time points, and the general pattern and distribution of compartments in the interaction matrices. As expected, we selected the first eigenvector as representative of compartment segregation for most of the chromosomes in all time points. However, in the cases of chromosomes 4 and 7, we selected their second eigenvectors, this again, for all time points. Following the accepted protocol, the selected eigenvectors were rotated according to their enrichment in markers for transcriptional activity in order to associate positive values to A-compartment and negative values to B-compartments. Finally, a sigmoid transformation was applied to the resulting eigenvector in order to reduce the impact of specific outliers across time points and chromosomes. We referred to the resulting transformed eigenvectors as compartment scores.

TAD borders were identified using the TADbit built-in TAD caller applied on 50 kb interaction matrices. TADbit's TAD caller assesses the robustness of the TAD border detection by assigning a score between 1 and 10 to each TAD border. In this work, we used only TAD borders with a score strictly above 4. As an alternative TAD calling strategy, we also used TAD-borders called using the insulation score as proposed in ref. 91. In our implementation, we measured levels of interactions in the distance range of 50 kb to 400 kb and considered a value of *delta = 2* (smoothing parameter, where 2 relates to the span in number of bins). This combination of parameters was found by maximizing the number of shared borders between replicates. TAD border alignments were generated with the function *align_TAD_borders* from our version of TADbit. We considered borders to be homologous between replicates if the distance between them was below or equal to 100 kb (2 bins).

### ChIP-seq processing
ENCODE standards were followed to process paired-end reads. Sequencing adapters were trimmed using Trim Galore! (0.6.6). Reads were then mapped to the reference genome (GRCh37.p13) using bowtie2 (2.3.2)[92] in the *--very-sensitive* mode. Low-quality reads, reads overlapping the ENCODE blacklist, and duplicate reads were filtered out using samtools (1.9)[93]. Genome-wide coverage was computed using the function bamCoverage from deepTools (3.2.1)[94] to obtain bigwig files for visualization purposes. Macs2 (2.2.7.1)[95] was used for peak calling in the narrow mode for H3K27ac and broad mode for H3K4me1, using an input sample as control, with default parameters. Consensus peaks were computed for each condition using Macs2 with all replicates and their respective input samples as control, setting the parameter *--scale-to small*. For quantification, a set of non-redundant enriched regions were defined by taking the union of peaks from all datasets. The signal of this set of regions was then quantified for each sample by counting the number of reads falling into each region using the function regionCounts from R package csaw (1.30.1)[96]. Normalization and differential analysis were performed using R package DESeq2 (1.36.0)[97]. Library size factors for normalization were calculated based on the background signal. This background signal was quantified per sample, excluding the set of previously defined non-redundant enriched regions. Specifically, this signal was quantified on fragment counts over genomic bins of 10 kb using the function windowCounts from R package csaw[96]. Library statistics were assessed using FastQC and MultiQC[98] and summarized in Suppl. Data 4.

### RNA-seq processing
ENCODE standards were followed to process paired-end reads. Sequencing adapters were trimmed using Trim Galore! (0.6.6). Reads were then mapped to the reference genome (GRCh37.p13) using STAR (2.7.0f)[99] with parameters recommended by ENCODE. Read counts were quantified using featureCounts from subread (2.0.0)[100]. Normalization and differential analysis were performed using R package DESeq2 (1.36.0)[97]. Library statistics were assessed using FastQC and MultiQC[98] and summarized in Suppl. Data 5.

### Definition of promoters and enhancers
Gene promoters were defined as regions spanning 1000 base pairs upstream and 200 base pairs downstream from their transcriptional start site. Ensembl gene annotation GRCh37 release 87 was used to define transcriptional start sites. To define enhancers, we first identified the intersection between consensus H3K4me1 and H3K27ac peaks at 0 h, 1 h and 10 h after p53 activation in wild-type conditions. Consensus enhancers were defined as the union of enhancers at each time point. To define enhancer activity, we annotated whether consensus H3K4me1 and H3K27ac peaks were present for each time point, with solely H3K4me1 presence signifying a primed enhancer state, and both H3K4me1 and H3K27ac presence signifying an active enhancer state. Defined enhancer regions used for downstream analysis are presented in Suppl. Data 6.

### Definition of functional p53 binding sites
Since p53 binding is stimulus and tissue independent[27,101,102], but its functionality displays clear cell type- and stimulus-specificity, we defined a list of functional p53 binding sites at 1 and 10 h of 10 μM Nutlin-3a treatment. Specifically, we used publicly available ChIP-seq data of p53 in the HCT116 cell line treated with 10 μM Nutlin-3a during 12 h from[26] (GSE86164). This dataset was refined into time point-specific functional p53 binding sites by intersecting it with our H3K27ac consensus peaks dataset. Specifically, the H3K27ac consensus peaks of 1 and 10 h time points in wild-type conditions were overlapped with the p53 binding sites using R package GenomicRanges (1.50.2)[103]. A p53 binding site was defined as functional when overlapping at least 1 base pair with an H3K27ac peak. The set of functional p53 binding sites are listed in Suppl. Data 7.

### PCHi-C processing
Raw sequencing reads were processed using HiCUP (0.8.2)[104]. The target sequence of the restriction enzyme HindIII was used to computationally digest the genome. HiCUP was then used to map paired-end reads to the human genome (GRCh37.p13), filter out experimental artifacts such as circularized reads and re-ligations, and remove duplicate reads. Paired reads which do not overlap with a captured restriction fragment are filtered out, retaining only uniquely captured valid reads for downstream analysis. This information is then used to assess the sample's capture efficiency. Datasets were scaled down to the same sequencing depth in reference to the corresponding biological replicate of the dataset with the lowest number of unique valid reads (the 1 h time point datasets). Interaction confidence scores were computed using the R package CHiCAGO (1.24.0)[105]. Interactions with a CHiCAGO score ≥5 were considered high-confidence interactions. CHiCAGO scores were recalibrated based on control datasets (0 h) using the fitDistCurve function from the R package CHiCAGO. Library statistics were assessed using FastQC and MultiQC[98] and are presented in Suppl. Data 8.

Linear generalized additive model (linear GAM) was used to fit percentages of observed significant interactions at different genomic distance brackets with splines. For all samples, the residuals calculated were all above 0.99 and the p values associated with the number of parameters below 1e-16, justifying the optimized (grid-search) choice of 21, 31, and 30 splines for 0 h, 1 h and 10 h time points, respectively (smoothing lambda parameter optimized together at 0.1 for all three samples). We used the GAM implementation from pyGAM[106]. Prediction intervals were calculated from the distribution of observed data

**Article** https://doi.org/10.1038/s41467-024-46666-1

points. Confidence intervals of the model, not shown, were fully included inside prediction intervals.

## Definition of p53 target genes

Proximal p53 target genes are defined as those genes with an overlapping functional p53 binding sites at their promoter. Distal p53 target genes are determined based on high-confidence interactions defined using CHiCAGO (1.24.0)[105] (CHiCAGO score ≥ 5). These distal genes must be located in captured fragment interacting with a fragment that overlaps a functional p53 binding site that i) is not defined as proximally targeting, and ii) overlaps an active enhancer. For each p53 binding site, a prioritization strategy was implemented whereby the distal target gene with the largest gain in H3K27ac at their promoter (based on the mean log2 fold change) was prioritized as their target gene. Genes targeted by the same p53 binding site and with the same mean log2 fold change were all defined equally as distal target genes. The set of p53 target genes is summarized in Suppl. Data 9.

## Gene set enrichment analysis

Gene set enrichment analysis (GSEA) was performed using the enricher function from clusterProfiler (4.4.4)[107] using default parameters. *p*-values were computed using a hypergeometric distribution test and adjusted for multiple comparisons using Benjamini-Hochberg correction. Gene sets were defined as enriched with an adjusted *p* value ≤ 0.05. Gene sets were obtained from the Molecular signatures database[107] via the msigdbr R package (7.5.1). Analysis was performed against all Human Collections, with particular emphasis on H collection (hallmark gene sets)[108], C2 collection (curated gene sets), C5 collection (ontology gene sets)[109–111] and C6 collection (oncogenic signature gene sets). Depending on the analysis, we tested given sets of genes against different libraries of total genes (often referred to as a gene universe). For the GSEA of RNA-seq and ChIP-seq differential analysis results, we used the Ensembl gene annotation GRCh37 release 87. For the GSEA of p53 target genes, we used genes present captured HindIII fragments. All GSEA results are presented in Suppl. Data 10.

## Statistical methods and figures

PCA of TAD borders and compartments was performed using the scikit-learn API v1.1.3. Statistical tests and complex numerical treatments were performed with SciPy v1.10.1[112] and NumPy v.1.24.2[113]. Matrix comparison and large internal data manipulation were performed using the Pandas API. Figures and plots were generated using ggplot2 (Wickham H (2016). *ggplot2: Elegant Graphics for Data Analysis.* Springer-Verlag New York. ISBN 978-3-319-24277-4) and matplotlib v3.7.1[114].

PCA for RNA-seq, ChIP-seq, and PCHi-C samples were generated using prcomp function from stats (4.2.1) R package[115]. Statistical tests were performed using ggpubr (0.6.0)[116]. Data manipulation and complex numerical treatments were performed using tidyverse (1.3.2).

Weighted Euclidean distances were applied on principal components (PCs) computed from Hi-C data to measure the similarity of biological replicates. The formula applied was:

$$WED_{ij} = \sqrt{\sum_{n=1}^{N} \left( PC_{ni} - PC_{nj} \right)^2 \times w_n} \qquad (1)$$

where *i* and *j* are biological replicates, *N* the number of considered principal components (10 here), and $w_n$ is the variance explained by the current PC, and $PC_{ni}$ the current (*n*) PC of the sample *i*.

For analysis regarding the identification of p53 target genes, mean comparisons were performed using a two-sided Mann-Whitney test. Data manipulation and integration with PCHi-C data sets were performed using HiCaptuRe[117].

We implemented the HiCRep algorithm[118] in TADbit to compute the stratum-adjusted correlation coefficient (SCC) between the two replicates.

## Data visualization

Data from this study can be visualized in the Washington University Epigenome Browser (http://epigenomegateway.wustl.edu/browser/) [http://epigenomegateway.wustl.edu/browser/].

## Reporting summary

Further information on research design is available in the Nature Portfolio Reporting Summary linked to this article.

## Data availability

Raw and processed sequencing data for Hi-C, ChIP-seq, RNA-seq and PCHi-C data generated in this study have been deposited in the Gene Expression Omnibus (GEO) database under accession code GSE235947. The nascent qRT-PCR, western blot quantification, functional p53 binding sites and enhancer data generated in this study are provided in the Supplementary Information. The p53 binding sites data used in this study are available in the Gene Expression Omnibus (GEO) database under accession code GSE86164 [https://www.ncbi.nlm.nih.gov/geo/query/acc.cgi]. Source data are provided with this paper.

## Code availability

Scripts to reproduce the analysis and figures in this study are available on GitHub https://github.com/JavierreLab/p53 [https://github.com/JavierreLab/p53] (https://doi.org/10.5281/zenodo.8075023) [https://zenodo.org/records/8075024].

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

## Acknowledgements

We gratefully acknowledge Masato T. Kanemaki and Sergi Cuartero for kindly sharing his HCT116 RAD21-mAID-mCloverCMV-OsTIR1(F74G) cell line model[44] and his antibody against RAD21 respectively. We thank Joaquin M. Espinosa team for critical discussion about publicly available RNA-seq datasets. Finally, we want also to thank Wouter de Laat and members of the Javierre Group for critical discussion. We thank CERCA Programme/Generalitat de Catalunya and the Josep Carreras Foundation for institutional support. This work was supported by FEDER/Spanish Ministry of Science and Innovation (PID2021-125277OB-I00 and PID2019-111243RA-I00), the European Hematology Association (4823998) and the Scientific Foundation of the Spanish Association Against Cancer (AECC) (LABAE21981JAVI). A.N.-A. is funded by the FEDER/Spanish Ministry of Science and Innovation (FPI fellowship PRE2022-102463), and A.L. is funded by Worldwide Cancer Research (20-0269), L.R. is funded by AGAUR (FI fellowship 2019FI-B00017), A.A.-M. is funded by the Spanish Association Against Cancer AECC (PRDBA233916ALCA), E.M.N. is funded by the European Union's Horizon 2020 research and innovation program under the Marie Skłodowska-Curie grant agreement No. 955534, S.D.C. is funded by la Caixa Banking Foundation Junior Leader project (LCF/BQ/PR20/11770002), J.L.S. is funded by the Spanish Health Institute Carlos III (CP19/00176) and B.M.J. is funded by the Spanish Health Institute Carlos III (CP22/00127). The funding bodies were not involved in the study design, collection, analysis, interpretation of data, the writing of this article or the decision to submit it for publication.

## Author contributions

F.S. and B.M.J. designed research and supervised the project; L.F.-E.., L.R., A.L., B.U., A.A.-M., E.M.N. and B.M.J. performed research; F.S., A.N.-A., M.C.-P. analyzed data; A.L.O., P.F., M.G., S.D.C., J.L.S., and A.V. provide critical feedback; B.M.J. and A.N.-A. wrote the paper and all authors provided feedback on the manuscript.

## Competing interests

The authors have no competing interests.
