## [Peer Review File · Nature Communications]

p53 rapidly restructures 3D chromatin organization to trigger a transcriptional responseREVIEWER COMMENTS

Reviewer #1 (Remarks to the Author):

Serra et al have generated HiC, PCHiC and histone mark ChIP data from various time points following p53 activation (via Nutlin3a) in HCT116 cell lines. They propose this leads to two sharp waves of genome organisational change, directly and indirectly regulated by p53 activation, impacting gene expression in a long-range and cohesin-dependent manner. I have a number of concerns about the work – minor and major – and would suggest significant changes to the current manuscript.

1. Fundamentally, I am unsurprised by the conclusion that activating a master transcription factor has an impact of 3D genome organisation rapidly and at all scales in a cohesin dependent manner. Numerous transcription factors have been shown to perform such functions - CTCF for example.

2. The manuscript currently contains many overstated conclusions and examples of inaccurate language. In the abstract alone, “within minutes of its activation” = 1hr, “340 genes directly regulated by p53 over distance, most of these not previously identified” = what distance? how many is most? There are too many of these to list here, but the authors must tone down their conclusions and work on linguistic accuracy throughout the work.

3. In my opinion the work needs to be more concise. I would propose the 7 figures could readily be trimmed to 3 major figures without any loss of narrative or accuracy. The bulk of figure panels are minimally discussed and add little to the narrative.

4. Could the authors strengthen their argument re the direct impacts of p53 on genome architecture by removing p53 after the 24hr time point to explore which of the reported changes are reversible. This could either be done chemically (removing Nutlin3a) or genetically (p53-AID)?

5. Fig 1 could be much more concise. 1B is should be moved to Supp Fig 1. Fig 1F, G and H show very similar information that doesn't really add significantly to the narrative.

6. Line 116-117 – why is the early gain of activity unexpected, given the requirement for rapid change after p53 induction?

7. Line 124 – “While 2,091 of TAD borders were stable across all stages, the remaining ones were dynamically lost or gained.” How many remaining?

8. Line 127 – “TAD profiles were then extensively conserved until the second wave of rewiring occurring after 10 hours of treatment”. The suggested two waves of architectural change needs better visualisation and evidence. For example, I would suggest showing only up and down changes in Fig 2B (not invariant and not variant number but only changes). I would also argue that 2C and F suggest a more gradual/continuous change in TAD boundaries across time (with the possible exception of between 1-4 hours). Currently I am unconvinced by the two waves of “sharp acquisition of TAD borders”.

9. Line 160 - “p53 binding to chromatin is largely invariant... Thus, we defined as functional p53 binding sites a set of 2,105 sites bound by p53 and characterized by the presence of the activating H3K27ac histone mark”. By the authors own statement these regions should be “active promoters”; however, they then go on to state “Only 6.5% (137/2,105) of those functional p53 bindings occurred at promoters”. How is this possible?

10. I believe Supp Fig 3G-I are mislabeled to all include H3K27ac. I suspect one set is actually H3K4me1.

11. This section is an example of another major concern regarding the subcategorization of genes or regions based upon poorly described and/or justified criteria (eg. Line 241-243 – 253 enhancers to 340 genes in “long range”. What is long range? Line 180-182 – if the authors define functional

p53 sites by the presence of p53 AND active promoters they be definition going to be enriched in active genomic regions eg. A compartments and TAD boundaries).

12. If my interpretation of the authors definition of functional p53 binding sites is correct then the conclusionary statement "Collectively, these results highlight p53's preference to bind active regulatory elements and suggest its limited capacity of epigenetic rewiring on the linear genome." is untrue.

13. While the more informative section of the work, the promoter interactome section needs better visualization. Currently it is not possible for a reader to draw their own conclusions on the data or the authors statements. For example, I would replace Fig 4C with an upset plot showing the relationship between all three time points. I would also like to see the data contained in Fig 4D better reflecting the conclusions. For example, either zoomed in on the regions discussed or numerical values of average PCHiC per distance range across particular ranges plotted as individual points to allow better comparisons between the samples.

14. I would either discuss the absence of detectable change in expression 1hr post-Nut3a or remove Fig 5G. I understand that Fig5K show nascent changes.

15. Given what appears to be differential impacts on expression of p53 activation and cohesin depletion, could the authors look at the relative reliance of candidate genes (Fig 5K) on long range regulation? For example in PCHiC data?

16. Did the authors check what impact cohesin depletion prior to Nut3a treatment had on gene expression?

Reviewer #2 (Remarks to the Author):

Serra et al. used state of the art genomics to describe the impact of p53 activation in 3D genome organization, that instructs changes in gene expression dictated by establishment of distal enhancer-promoter interactions. While studies with a similar scope were published in the past (i.e. PMID 27741251, 33823140 and others), this work provide a valuable, time-lapsed high-resolution map of genome rewiring following p53 activation the goes beyond published work. This allows a better understanding of bona fide p53 targets and, or a larger perspective, provides a model to understand genome plasticity, the role of 3D genome architecture and particularly, the role of distal promoter-enhancer interaction.

Data and methods are sound, statements are supported by experimental evidences.

Minor points:

line 45: being -one of - the most studied genes in history

line 49: at megabase-scale

line 55: please provide a reference about TADs being conserved across species

line 68: pushes it rather than pumps it

line 97: Suppl Fig 1A. Are authors using equal loading? WB of nuclear p53 would be informative of activation.

Major points:

It would be beneficial to attempt a p53 Promoter-capture Hi-ChIP experiment (at the most informative time point). This would allow to directly link the role and involvement of p53 in the establishment of loops (otherwise based on publicly available p53 chip-seq data).

Reviewer #3 (Remarks to the Author):

Serra et al. investigate changes in 3D genome conformation in a colon cancer cell line in which the p53 stress response was pharmacologically activated. The authors generated several omics datasets across a time series, from which they draw associations aimed at describing the relationship between p53 pathway-activated transcriptional changes and 3D genome conformation. Technically, the presented data appear of high quality and suitable for answering the research question at hand. Analyses largely appear appropriate, although at occasions superficial. However, I feel that the data is often overinterpreted and conclusions drawn too strong, especially considering a general lack of functional experiments to validate new p53 target genes/enhancers or no assessment of 3D genome conformation in Cohesin-depleted cells in which the p53 pathway was stimulated. As such, I think the paper falls short in delivering what is promised in the manuscript's rather boldly written abstract.

Specific comments:

- I feel that the authors should clearly mention in their Introduction that there have been reports linking p53 binding at enhancers to changes in 3D genome folding. Some of these papers are cited (i.e. refs 20-22), but deserve to be briefly discussed.
- The data in Supp.Fig.1A-C contains relevant QC information that are currently not used in the paper. The authors should discuss these data.
- Fig.2F: I'm not sure how the authors can conclude from this graph that 'less insulating borders are preferentially erased during p53 activation'. This would require separately showing insulation scores at lost borders vs. stable (or gained) borders.
- The authors use previously generated p53 ChIP-Seq data, although timepoints do not entirely match between studies it seems (e.g. 10h vs 12h)? The authors focus quite some of their analyses on their 10h timepoint, but does that mean that they used 12h p53 binding data? If so, this needs to be clearly stated.
- I find Fig.3D-E very difficult to read and understand; I strongly encourage the authors to use a simpler way of visualizing the enrichment of p53 binding in the different switching categories.
- Fig.2/3: why don't the authors attempt to connect TAD dynamics to transcriptional changes induced by p53 pathway activation, or p53 binding dynamics? This seems a very relevant angle to pursue.
- Line 198: 'solely' is a very strong word, the authors cannot exclude that examples exists were p53 represses or induced A->B switching, so I suggest using 'mainly' instead.
- Fig.5G shows no transcriptional changes in the p53 target genes identified after 1h of activation, despite increases in p53 response element H3K27Ac levels. It seems at odds with p53-mediated topological/epigenomic changes being relevant for the acute transcriptional reprogramming induced by p53 pathway activation. In panel K, the authors show nascent RNA levels for some genes increasing at 1h, but in my opinion one still has to conclude that the majority of the immediate p53 target genes are not upregulated after 1h.
- Fig.6. How do these two different scenarios (i.e. pre-established vs de novo) relate to the transcriptional changes at these target genes? This is important to investigate, as the authors claim that p53-mediated topological changes are underlying the transcriptional changes observed upon nutlin treatment.
- Fig.7D. Most p53 target genes do not seem much affected by the Cohesin depletion, especially at the 1h timepoint (2 out of 10). This doesn't seem like a firm basis for concluding that 'Rad21 degradation compromises p53-driven transcription responses'.
- Data presented in Fig.S6C seems to show a more convincing transcriptional impact of Rad21 depletion on p53 target genes. How was this data generated (qPCR as well)? And why is it different from the data shown in 7D?
- The lack of (PC)Hi-C data on the Rad21-depleted nutlin-treated cells precludes any conclusions on whether p53-mediated topological changes are cohesin-dependent, such as those stated in the abstract: 'Finally, we showcase that p53 controls transcription of distal genes through newly formed and pre-existing enhancer-promoter loops in a cohesin dependent manner.'
- Overall, the authors state that one of the key strengths of this paper is that it provides a reliable set of new p53 target genes. However, these genes might be very specific to the colon carcinoma cell line used. How do the 340 supposedly direct target genes - which the authors only based on a measured PChi-C interaction (see comment below) - hold up in other cell systems in which the p56

pathway is activated?

- I find the evidence presented for 'a reliable set of new p53 direct target genes' (quote from their abstract) rather underwhelming. Sure, these genes are linked in 3D to a p53-bound enhancer, but that is no formal proof that these enhancers actually regulate these genes and that p53 is critical for the enhancing effect of said element. Since the authors use a cell line model that is amenable to genome editing (as shown by Fig.7), I would have expected the authors to use CRISPR-Cas9 technology to experimentally validate at least a few of their hypothesized enhancer-promoter links by cutting out the p52 enhancer - or even better disrupt the p53 binding motif in the enhancer. It is this type of proof that the authors in my opinion need to make a compelling case for a novel set of reliable p53 target genes - especially if as stated in the discussion most of these new target genes have not been detected by the dozens of earlier studies conducted.

Point-by-point responses to Reviewers' comments (NCOMMS-23-26694-T).

First, we would like to thank the reviewers for the constructive criticism and suggestions to improve the manuscript. During revision, we have included the following key improvements:

- Functional validation of p53 distal target genes using CRISPR-based deletion of p53 binding sites at enhancers, as suggested by reviewer #3.
- Experimental confirmation of DNA loops' erasure upon RAD21 depletion applying PChi-C methodology to analyze the degron model, as suggested by reviewers #1 and #3.
- Study of the reversibility of the p53-associated topological alterations by generating Hi-C libraries after p53 inactivation via wash of Nutlin-3a, as suggested by reviewer #1.
- WB-based analysis of nuclear extracts, rather than whole protein extracts, to quantify abundance of p53 and p53-regulated proteins, as suggested by reviewer #2.
- Transcriptional validation of additional candidate genes to be distally regulated by p53-bound enhancers. Besides, we have increased the number of biological replicates to increase the statistical power to ultimately detect additional significant differences at the nascent transcription level, as suggested by reviewer #3.
- Investigation of the cell type specificity of the set of genes transcriptionally controlled by p53-bound enhancers, as suggested by reviewer #3.
- Correction of inaccurate language or conclusions; inclusion of clarifications and additional data, and reduction of main figures to ultimately improve the quality of the manuscript, as suggested by reviewer #1.

To help with the structuring of this document, we have marked the reviewer's comments with **green typography**, while our replies are shown in normal black font. Besides, we have tracked changes on the manuscript to identify text improvements more easily.

Reviewer #1 (Remarks to the Author):

Serra et al have generated HiC, PCHiC and histone mark ChIP data from various time points following p53 activation (via Nutlin-3a) in HCT116 cell lines. They propose this leads to two sharp waves of genome organisational change, directly and indirectly regulated by p53 activation, impacting gene expression in a long-range and cohesin-dependent manner. I have a number of concerns about the work – minor and major – and would suggest significant changes to the current manuscript.

We thank the reviewer for the constructive suggestions to improve the manuscript.

Comments:

1. Fundamentally, I am unsurprised by the conclusion that activating a master transcription factor has an impact on 3D genome organisation rapidly and at all scales in a cohesin dependent manner. Numerous transcription factors have been shown to perform such functions - CTCF for example.

We agree that there are some examples of transcription factors that control 3D chromatin organization. However, it is important to highlight that p53 was not among these transcription factors described as able to modulate the 3D genome organization. Moreover, most of these studies lack the resolution to detect the speed of 3D chromatin rewiring since these have not performed time-lapsed analyses (*i.e.*, most of them analyse the 3D chromatin organization days after genetic perturbation) (*e.g.*, Hu et al, Cell 2023 a).

Specifically, three studies (Melo et al, Mol Cell 2013; Link et al, Genes Dev 2013; Millau et al, Plos One 2016) were performed to decipher the interplay between p53 and genome architecture, agreeing that p53-engaged DNA loops represented pre-existing chromatin architectures. However, these conclusions cannot be generalized as only a few p53-bound elements or p53 target genes were analysed. In our manuscript we report a time-lapsed analysis that decodes, for the first time, the 3D genome organization rewiring associated with p53 activation, which occurs very early upon cellular perturbation (*i.e.*, within one hour of p53 activation). Using a genome-wide approach, we demonstrate that p53 relies not only on pre-established DNA loops, but also on the generation of new DNA loops to connect p53-bound enhancers with their distal target promoters to control gene transcription. To summarize, our

results provide novel mechanistic insights on p53's role as an architectural regulator at distinct topological layers, as well as a new perspective on the dynamism of the 3D genome, and the speed at which this can be achieved after being triggered by a transcription factor.

2. The manuscript currently contains many overstated conclusions and examples of inaccurate language. In the abstract alone, "within minutes of its activation" = 1hr, "340 genes directly regulated by p53 over distance, most of these not previously identified" = what distance? How many is most? There are too many of these to list here, but the authors must tone down their conclusions and work on linguistic accuracy throughout the work.

We apologize for the presence of examples of inaccurate language or overstated conclusions. We have tried to correct these throughout the manuscript.

3. In my opinion the work needs to be more concise. I would propose the 7 figures could readily be trimmed to 3 major figures without any loss of narrative or accuracy. The bulk of figure panels are minimally discussed and add little to the narrative.

We thank the referee for the feedback. Nature Communication allows up to 10 main display items (figures and/or tables). Moreover, supplementary figures are not easily accessible as they form part of the supplementary information that can be downloaded in PDF format from a link. For these reasons, advice from the editor would be very valuable to decide the final number of main figures to ultimately fulfil your requirement without losing information. In any case, to be more concise, we have moved panels 1B, 1E, 1H, 3B, 4A, 4E, 7A and 7B to supplementary figures and have elaborated the discussion of some of these panels in the main text. In addition, panels 3D and 3E have been removed from the manuscript.

4. Could the authors strengthen their argument re the direct impacts of p53 on genome architecture by removing p53 after the 24hr time point to explore which of the reported changes are reversible. This could either be done chemically (removing Nutlin3-a) or genetically (p53-AID)?

We completely agree that investigating the reversibility of the p53-driven changes may significantly improve the quality of the work, and we thank the referee for this elegant suggestion.

To address this point, we optimized the treatment conditions to recover basal conditions prior to Nutlin-3a treatment. To do so, we treated HCT116 cells with 10 μ M Nutlin-3a or the drug vehicle (DMSO). After 24 hours, cells were washed twice with PBS to remove Nutlin-3a or DMSO and cultured in fresh growth media without Nutlin-3a for an additional 24 or 48 hours. Besides, cells were collected before washing Nutlin-3a as reference of p53 activation.

According to WB analysis (**Fig. A** and Suppl. Fig. 1 D-G in the revised manuscript), p53 protein abundance and p53 activation levels (according to phosphorylation of serin 46) progressively increased during the 24 hours after Nutlin-3a treatment and decreased after Nutlin-3a removal, reaching basal conditions (*i.e.*, conditions exhibited by cells treated with the drug vehicle) 24 hours after Nutlin-3a removal.

Fig A. Western blot analysis of nuclear and cytosolic p53, p21 and activated p53 (according to phosphorylation of serin 46; p53 (pSer46)) levels using Vinculin and histone 3 (H3) as controls. Nut 0h (cells collected after 24 hours of treatment with drug vehicle), Nut 1h, Nut 10h and Nut 24h (cells collected 1, 10 and 24 hours after treatment with 10 μ M Nutlin-3a, respectively); Wash 24h, Wash 48h (cells cultured for 24 hours after treatment with 10 μ M Nutlin-3a, followed by 24 or 48 additional hours without Nutlin-3a).

Analysis of nascent transcript levels (**Fig. B** and Suppl. Fig. 1 B-C) and protein abundance (**Fig. A** and Suppl. Fig. 1 D-G) of the well-known p53 target genes (*i.e.*, *CDKN1A/p21*, *TGFA* and *BAX*) suggested that, although basal conditions of p53 activation level were recovered 24 hours after Nutlin-3a removal, the p53 driven transcriptional changes were not fully recovered. For this reason, we selected a combinatory treatment composed of 24 hours of cell culture after 10 μ M Nutlin-3a followed by 24 additional hours in fresh growth media without Nutlin-3a to study the reversibility of the p53-driven changes and investigate whether these are direct p53 effects (*i.e.*, changes provoked by p53 binding that should be reverted after recovering basal p53 levels) or indirect downstream effects of p53 pathway's activation that may or may not be recovered.

Fig. B. Relative nascent mRNA expression (fold change) of well-known p53 targets *CDKN1A*, *TGFA*, and *BAX* along p53 activation and inactivation. Relative expression was averaged between the replicates at each time point for each gene. Cells were collected 24 hours after treatment with 10 μ M Nutlin-3a (labelled as *Nut 24h*) or drug vehicle (DMSO; labelled as *Nut 0h*), washed and cultured for another 0, 24 and 48 hours with fresh growth media without Nutlin-3a (labelled as *No Wash*, *Wash 24h*, and *Wash 48h*, respectively).

Next, we used *in situ* Hi-C and fixed chromatin to generate genome-wide chromosome conformation maps (Suppl. Data 3). After confirming libraries quality and reproducibility between biological replicates (SCC > 0.95; see methods) (Suppl. Fig. 1H-I), we segmented the genome into A and B compartments (Fig. 1B-E and Suppl. Fig. 1J-O) and we used chromosome-wide insulation potential (TADbit score > 4) to identify TAD borders (Fig. 2 and Suppl. Fig. 2A-D) as we previously did for the analysis of the Hi-C time-course along p53 activation. Globally, we observed that a significant proportion of topological changes occurring throughout p53 activation were reverted, especially changes associated with the second wave of rewiring, clearly observed at the TAD level and, at less extend, at the compartment level.

Collectively, these results demonstrate dramatic and reversible changes in genome architecture throughout p53 activation and inactivation and support the capacity of p53 to drive direct and indirect changes in genome architecture.

All these new findings have been incorporated into the manuscript (Fig. 1 and 2, Suppl. Fig. 1 and 2).

5. Fig 1 could be much more concise. 1B should be moved to Supp Fig 1. Fig 1F, G and H show very similar information that doesn't really add significantly to the narrative.

To make Fig.1 more concise, we have moved panels 1B, 1E and 1H to Suppl. Fig. 1. For more details, please see point 3.

6. Line 116-117 – why is the early gain of activity unexpected, given the requirement for rapid change after p53 induction?

There are two main reasons why we did not expect such a dramatic rewiring of the 3D chromatin organization upon only one hour of p53 activation. First, with a few exceptions, most genome-wide alterations in 3D genome architecture associated with any cellular perturbation have been reported at later time points, an experimental design of time-resolved studies suggesting a slower chromatin dynamism. Second, according to both literature and our findings, after 1 hour of p53 activation there is a modest accumulation and activation of the p53 protein, a low transcriptional activation of well-known p53 target genes, and an absence of significant cell cycle alterations or apoptosis when compared with longer treatments (*e.g.*, 10 hours of p53 activation) (Suppl. Fig. 1 A-G). In consequence, we expected to detect significant topological alterations mainly at later time points of the time course.

We have tried to highlight both reasons in the discussion to support our claim of “unexpected changes”.

7. Line 124 – “While 2,091 of TAD borders were stable across all stages, the remaining ones were dynamically lost or gained.” How many remaining?

The number of timepoint specific (variant) TAD borders is detailed in Fig. 2B (dark grey part of the bar plot). To avoid misunderstanding, we have included this information in the figure legend, and we have provided the number of total TAD borders detected across all time points (4610) in the main text. Note that number of invariant and total TADs borders have slightly changed after including in the analysis the samples generated after washing Nutlin-3a and culturing cells for 24 additional hours in fresh growth media without Nutlin-3a.

8. Line 127 – “TAD profiles were then extensively conserved until the second wave of rewiring occurred after 10 hours of treatment”. The suggested two waves of architectural change need better visualisation and evidence. For example, I would suggest showing only up and down changes in Fig 2B (not invariant and not variant number but only changes). I would also argue that 2C and F suggest a more gradual/continuous change in TAD boundaries across time (with

the possible exception of between 1-4 hours). Currently I am unconvinced by the two waves of “sharp acquisition of TAD borders”.

To improve the visualization of TADs dynamism we have improved Fig. 2B (pie charts represent the number of gained and lost TADs borders in each transition) and Fig. 2D (by highlighting and associating the more informative clusters that support both waves of rewiring). Both modifications help to visualize these waves that engage different TAD borders depending on Nutlin-3a exposure. Indeed, both waves can also be identified by PCA (Fig. 2C), where PC2 captures the first wave occurring at 1 hour of p53 activation and PC1 reflects the second wave occurring at 10 hours Nutlin-3a treatment and onwards. Besides, to better respond to the referee’s observation, we have modified the text by also claiming a non-dynamic period between 1 and 7 hours of p53 activation and we have removed the adjective “sharp”.

9. Line 160 - “p53 binding to chromatin is largely invariant... Thus, we identified as functional p53 binding sites a set of 2,105 sites bound by p53 and characterised by the presence of the activating H3K27ac histone mark”. By the authors own statement these regions should be “active promoters”; however, they then go on to state “Only 6.5% (137/2,105) of those functional p53 bindings occurred at promoters”. How is this possible?

We apologize for the misunderstanding. The H3K27ac histone mark is not only associated with active promoters, but it is also associated with active enhancers. Therefore, those 2,105 p53 binding sites co-localizing with a H3K27ac peak under p53 activation can be identified as active promoters, active enhancers or unknown elements. By using the definition of promoters extensively used by the scientific community (-1,000/+200 nt from the transcriptional start site (TSS)), and ensembl gene annotation GRCh37 to define said TSS, only 137 out of 2,105 of p53 binding events occur at promoter regions. Of those remaining, 901 overlap with active enhancers defined throughout p53 activation (*i.e.*, co-localization for H3K27ac and H3K4me1, see methods). The final 1,067 p53 binding events co-localizing with a H3K27ac peak during p53 activation did not overlap with a defined promoter or enhancer region, for which reason they were classified as unknown. All these numbers have been included in Suppl. Fig. 4G.

10. I believe Supp Fig 3G-I are mislabeled to all include H3K27ac. I suspect one set is actually H3K4me1.

Please, let us clarify this misunderstanding. Suppl. Fig. 3H (now Suppl. Fig. 4H in the revised version of the manuscript) refers to H3K27ac levels at p53-bound promoters, Suppl. Fig. 3J (now Suppl. Fig. 4J in the revised version of the manuscript) refers to H3K27ac levels at p53-bound enhancers and Suppl. Fig. 3I (now Suppl. Fig. 4I in the revised version of the manuscript) refers to expression levels of genes characterized by p53-bound promoters.

We have tried to better clarify this in the figure legend.

11. This section is an example of another major concern regarding the subcategorization of genes or regions based upon poorly described and/or justified criteria (eg. Line 241-243 – 253 enhancers to 340 genes in “long range”. What is long range? Line 180-182 – if the authors define functional p53 sites by the presence of p53 AND active promoters they be definition going to be enriched in active genomic regions eg. A compartments and TAD boundaries).

As previously explained in point number 9, we identified functional p53 binding events (*i.e.*, p53 binding events overlapping H3K27ac peaks identified during p53 activation, which may be depleted of H3K27ac before Nutlin-3a treatment). These functional p53 binding events were classified as either: i) a promoter binding event (137 out of 2,105 p53 functional p53 binding sites) when overlapping a gene promoter region (*i.e.*, -1000/+200 nucleotides from the TSS) or ii) an enhancer binding event (901 out of 2,105 p53 functional p53 binding sites) when overlapping an enhancer, defined as described in the methods section (Fig. Suppl. 4G; Fig. Suppl. 3G of the old version of the manuscript, see methods). The remaining ones were classified as “unknown”.

Regarding functional p53 binding events at promoters (named as “promoter” in Fig. 5E and H), the associated genes were the gene regulated by the given promoter harboring the functional p53 binding site. These genes were labelled as “proximal” genes (Fig. 5B, D, F, G, I and J) and the distance between the functional p53 binding and the TSS is below 1000 nucleotides, given the use of the definition of promoters extensively used by the scientific community (-1,000/+200 nt from the TSS).

Regarding functional p53 binding events at enhancers (named as “enhancer” in Fig. 5E and H), the associated genes were identified using long-range chromatin interactions from PCHi-C data. These genes were labelled as “distal” genes (Fig. 5B, D, F, G, I and J). The median distance between the functional p53 binding and the TSS of its target gene is 116 kilobases. These genes (340 in total) are distally (*i.e.*, over distance, through DNA loops) controlled by p53. Besides, we identified the linearly closest TSS to each functional p53 binding event at enhancer and we labelled these as “nearest” genes. Our analysis suggested that in most of the cases these “nearest” genes are not the target gene since these are not in physical proximity and are not upregulated upon p53 activation (please note that only 7.1% of p53-bound enhancer’s “distal” genes were also identified as their “nearest” gene).

To avoid misunderstandings, we have provided more details in the text and in the legend of Fig. 5).

12. If my interpretation of the authors definition of functional p53 binding sites is correct then the conclusionary statement “Collectively, these results highlight p53’s preference to bind active regulatory elements and suggest its limited capacity of epigenetic rewiring on the linear genome.” is untrue.

To clarify, we defined functional p53 binding events as p53 binding events overlapping H3K27ac peaks identified during p53 activation (after 1 or 10 hours of Nutlin-3a treatment). These genomic regions defined as functional p53 binding events may be or not depleted of H3K27ac before Nutlin-3a treatment, but in most of the cases these already have H3K27ac deposition at basal conditions.

Indeed, as we showed in Suppl. Fig. 4G-J, (Suppl. Fig. 3G-J of the old version of the manuscript), p53 preferentially binds active enhancers (*i.e.*, active enhancer defined as overlapping H3K27ac and H3K4me1 peaks, see methods for enhancer definition) and active promoters (*i.e.*, a promoter region, defined as -1,000/+200 nt from the TSS, overlapping an H3K27ac peak) both of which tend to already be active before p53 activation. p53 binding at these active promoters and enhancers after p53 activation via Nutlin-3a treatment is associated with an increase of H3K27ac levels, demonstrating an increase in activity of these regulatory elements. However, since in most cases these regulatory elements were already active before

p53 activation and there are few cases of *de novo* establishment of enhancers or transition from inactive to active promoters, we believe that p53 does not have capacity to epigenetically rewire the linear genome.

13. While the more informative section of the work, the promoter interactome section needs better visualisation. Currently it is not possible for a reader to draw their own conclusions on the data or the authors statements. For example, I would replace Fig 4C with an upset plot showing the relationship between all three time points. I would also like to see the data contained in Fig 4D better reflecting the conclusions. For example, either zoomed in on the regions discussed or numerical values of average PCHiC per distance range across particular ranges plotted as individual points to allow better comparisons between the samples.

We are aware of the caveats of using Venn diagrams and appreciate the suggestion of replacing this with an upset plot. However, from our point of view, the upset plot depicted in **Fig. C**: i) hides the proportions of lost and gained interactions that we show in the Venn diagrams that we really want to highlight, which are mentioned in the main text; and ii) shows comparisons that we do not need. Since this is our impression, and we may be wrong, we have maintained Fig. 4C (now Fig. 4B) as is. However, this may be replaced by **Fig. C** if deemed necessary.

Fig. C. Upset plot displaying the overlap between significant interactions across p53 activation.

Regarding Fig. 4D (now Fig. 4C in the revised manuscript), it is important to highlight that it is complemented by Fig. 4E (now Suppl. Fig. 5E in the revised version of the manuscript). Specifically, panel D shows the distance distribution and E shows statistical significance of the differences between these distance distributions represented in D by distance ranges. However, we agree that a zoom in on the region of interest may be also informative, so we have modified Fig. 4D (now Fig. 4C in the revised manuscript) to include a zoom in around the discussed

region. Finally, we have generated a table with the number of PCHi-C interactions per distance range (Suppl. Data 8).

14. I would either discuss the absence of detectable change in expression 1 hour post-Nut3a or remove Fig 5G. I understand that Fig5K shows nascent changes.

We have further discussed the absence of detectable changes upon 1 hour of activation with Nutlin-3a using RNA-seq data. We have also now made it clearer in the text that Fig. 5G refers to RNA-seq data and that Fig. 5K refers to nascent RNA expression, measured by qRT-PCR.

Regarding the RNA-seq data, there are two reasons that may explain its limitation in detecting transcriptional upregulation after 1 hour of p53 activation:

- As opposed to after 10 hours of Nutlin-3a treatment, a 1 hour treatment generally triggers lower activation, both according to H3K27ac levels (at p53-bound promoters, p53-bound enhancers and promoters of genes distally regulated by p53-bound enhancers) (Fig.5 E-J) and nascent transcriptional output (Fig. 5K). Besides, this is also supported by the limited activation and accumulation of the p53 protein (Suppl. Fig. 1D-G) and the moderate upregulation of well-known p53 target genes (*e.g.*, *CDKN1A/p21*, *TGFA*, *BAX*) (Suppl. Fig. 1B, F, G)

- p53 activation leads to transcriptional changes of miRNAs and RNA binding proteins (RBPs) after 1 hour of its activation. Indeed, p53 has been identified as a major direct regulator of miRNAs (Feng.et al.). Consequently, levels of mRNA (detected by RNA-seq) may not capture the slight increase of transcriptional rates detected by qRT-PCR analysis, since miRNAs and RBPs may affect mRNA stability and ultimately alter the steady state levels of mature RNAs. Specifically, 7 and 4 microRNAs and 84 and 36 RBPs are significantly upregulated and downregulated, respectively, after 1 hour of p53 activation (Fig. D). Among these, 2 microRNAs (miR-1293 and miR-1915) and 9 RBPs (*SRSF1*, *EIF3*, *BRBM4*, *RBM14*, *ESRP2*, *RBM12B*, *PUF60*, *DNAJC17* and *EIF3B*) are distally controlled by p53-bound enhancers. Moreover, miR-4738 has a functional p53 binding at its promoter region.

Fig. D Differential expression analysis between 1 hour (Nut 1h) and 0 hour (Nut 0h) time points. Differentially expressed genes are those with log2 fold change (\log_2FC) $\neq 0$ and adjusted p value ≤ 0.05 (horizontal dotted line). Differentially expressed microRNAs (MIR) and RNA binding proteins (RBP) are highlighted in green or blue, respectively.

Furthermore, the limited capacity to identify early p53-driven transcriptional response quantifying steady state levels of mRNA was previously described by Allen et al.. Specifically, by performing comparative analysis of nascent RNA levels by GRO-seq (Core et al.) vs. RNA steady state levels by microarray, they showed that many p53 target genes transcribed at low levels are missed by microarray. For instance, *CDKN1A* and *TP53I3* mRNAs were not significantly upregulated after 1 hour of 10 μ M Nutlin-3a treatment, but their nascent transcription levels were found significantly increased. Interestingly, the GRO-seq data generated treating HCT116 cells with 10 μ M Nutlin-3a treatment identified 5 miRNAs directly transactivated by p53 (miR-1204, miR-3189, miR-34a, miR4679-1 and miR-4692). Besides, it is worth highlighting that nascent transcripts represent around 0.5% of total RNA content in a cell.

Finally, we also compared this publicly available GRO-seq data with our RNA-seq data at 1h and 10h after Nutlin-3a treatment. In fact, we identify only 40/198 (20%) to be significantly upregulated in our RNA-seq data after 1h hour of treatment, whereas 144/198 (73%,) of the genes identified by GRO-seq are found significantly upregulated in our RNA-seq data 10 hours after treatment (**Fig. E**). Furthermore, out of the 40 genes identified at 1 hour, 39 are also identified at 10 hours. Altogether, this clearly reflects the fact that RNA-seq cannot capture the complete picture of the transcriptional effects occurring only 1 hour after p53 activation, given that this technique is used to measure the global mRNA abundance, which is most likely affected by the regulation of splicing events.

Fig. E Upset plot depicting the overlap between genes found upregulated in publicly available GRO-seq data (which measured nascent RNA levels in HCT116 cells after 30 minutes of p53 activation by Nutlin-3a), and genes found significantly upregulated by RNA-seq after 1 and 1 hours of p53 activation by Nutlin-3a in our data, named as DEG_Nut1h and DEG_Nut10h, respectively.

Summarizing, our data, supported by the literature, demonstrates that the genes that we identified as distal targets of p53-bound enhancers are globally upregulated after 1 hour of p53 activation, albeit at low levels, and these cannot be detected by measuring mRNA abundance.

15. Given what appears to be differential impacts on expression of p53 activation and cohesin depletion, could the authors look at the relative reliance of candidate genes (Fig 5K) on long range regulation? For example in PCHiC data?

To demonstrate the reliance of these candidate genes on long-range interactions we have performed PCHi-C after RAD21 depletion (Suppl. Data 8 and Suppl. Fig. 9C-D). In accordance with previous findings (Thiecke *et al.*), most of the DNA loops engaging p53-bound enhancers and candidate target genes are erased (67%) or reduce their interaction strength (76%) upon cohesin depletion (Fig. 7A-B, G-H). The shorter-range DNA loops are less dependent on RAD21 depletion as previously reported (Rinzema *et al.*) (Fig. 7C, G-H).

Finally, we have performed CRISPR validation of 2 selected examples of distal target genes regulated by p53-bound enhancers, *S100A1* and *PCMI* genes (Fig. 7 G-I and Suppl. Fig. 9G-I). Specifically, we have performed two different deletions of the same p53 binding site found at an enhancer distally targeting *S100A1*, and two clones of the same deletion of a p53 binding site found at an enhancer distally targeting *PCMI* (4 homozygotic deletions in total). Indeed, these deletions prevent transcriptional upregulation of candidate distal target genes upon p53 activation (Fig. 7I). These results, in combination with a global increase in transcription levels

as well as an increase in H3K27ac deposition at their promoters upon p53 activation (Fig. 7D-F), support the reliance of our candidate distal target genes.

16. Did the authors check what impact cohesin depletion prior to Nut3a treatment had on gene expression?

To address referees' request, we have also checked the impact of RAD21 depletion prior to Nutlin-3a treatment on gene expression (Fig. F). To do this, we have normalized all our qRT-PCR data to the Nut 0h timepoint in WT condition (*i.e.*, treatment with drug vehicle DMSO and presence of RAD21) to factor in the effect of RAD21 depletion. Note that in the original Fig. 7D (now Fig. 7F in the revised manuscript), we normalized the 1 and 10 hours timepoints to the 0 hour time point of their respective condition (either wild type, WT, or RAD21 depletion, KD), to consider only the effect of p53 activation on expression.

Overall, we observe that, although with some exceptions (*i.e.*, *LAMC1* and *TENT4B*), cohesin depletion slightly downregulates the expression of our genes of interest, (yellow and light grey box plot in Fig. F and yellow and light grey bars in Fig. G), which can easily be associated with the loss of interactions between the gene and their distal enhancers. However, it is important to note that whilst we do see a downregulation of expression when removing cohesin before p53 activation, said downregulation is not statistically significant (Fig. F, $t(55.5) = 1.86$, $p = 0.0680$). Therefore, our initial observation still stands, since the downregulation as a result of RAD21 depletion after p53 activation is greater than before activation (green and dark grey box plot in Fig. F and green and dark grey bars in Fig. G, $t(4.36) = 38.0$, $p = 9.62e-05$). Consequently, the upregulation of these genes upon p53 activation in the presence of cohesin continues to be much greater than in cohesin depleted conditions, showcasing that p53 is dependent on these chromatin loops formed by cohesin to distally upregulate its target genes.

Fig. F. Distribution of nascent mRNA expression levels changes (fold change) of p53 distal target genes (*i.e.*, genes controlled by p53-bound enhancers), normalized to wild-type cells treated with drug vehicle (WT Nut 0h), and therefore considering the effects of RAD21 depletion as well as p53 activation before and after 1 hour of p53 activation. Wild type HCT116 cells (WT); RAD21 depleted HCT116 cells (KD).

Fig G. Levels of relative nascent mRNA expression (normalized to wild-type cells treated with drug vehicle, WT Nut 0h) of p53 distal target genes upon 1 hour of p53 activation in the presence (WT) or absence of RAD21 (KD)

Reviewer #2 (Remarks to the Author):

Serra *et al.*, used state of the art genomics to describe the impact of p53 activation in 3D genome organisation, that instructs changes in gene expression dictated by establishment of distal enhancer-promoter interactions. While studies with a similar scope were published in the past (i.e. PMID 27741251, 33823140 and others), this work provides a valuable, time-lapsed high-resolution map of genome rewiring following p53 activation that goes beyond published work. This allows a better understanding of bona fide p53 targets and, or a larger perspective, provides a model to understand genome plasticity, the role of 3D genome architecture and particularly, the role of distal promoter-enhancer interaction.

Data and methods are sound, statements are supported by experimental evidence.

We thank the reviewer for considering our work mechanically sound and novel. Besides, we appreciate the minor points, which have been addressed in the manuscript.

Minor points:

line 45: being -one of - the most studied genes in history

line 49: at megabase-scale

line 55: please provide a reference about TADs being conserved across species

line 68: pushes it rather than pumps it

line 97: Suppl Fig 1A. Are authors using equal loading? WB of nuclear p53 would be informative of activation.

We have extended and improved our Western blot (WB) analysis to validate p53 activation (Suppl. Fig. 1D-G). Specifically, we have quantified total p53 abundance of nuclear and cytosolic fractions along p53 activation and further inactivation*. Besides, we have quantified the abundance of an activated form of p53 characterized by the phosphorylation for serin 46

(pSer46) and p21 (encoded by the well-known p53 target gene *CDKN1A*). Histone 3 (H3) and Vinculin were used as loading controls for the analysis of the nuclear and cytosolic fractions respectively. Immunoblots were quantified to better compare samples independently of slight differences in sample loading (Suppl. Fig. 1D-G).

As clearly demonstrated by WB analysis and supported by quantification of nascent transcripts (Suppl. Fig. 1B-C), we observed a significant and progressive activation of p53 and its transcriptional response along the 24 hours of treatment using Nutlin-3a and inactivation after washing of Nutlin-3a.

* Please note that following a suggestion from reviewer #1, in the new version of the manuscript we have included an analysis on the reversibility of the p53-driven changes by removing Nutlin-3a after 24 hours of activation by washing with PBS and culturing cells for additions 24 hours with fresh growth media without Nutlin-3a).

Major points:

It would be beneficial to attempt a p53 Promoter-capture Hi-ChIP experiment (at the most informative time point). This would allow to directly link the role and involvement of p53 in the establishment of loops (otherwise based on publicly available p53 chip-seq data).

We completely agree on the added value of Hi-ChIP and we thank the reviewer for the suggestion. Indeed, Hi-ChIP was our first methodological choice. However, after several failed attempts due to the lack of a high-quality Hi-ChIP antibody against p53, we decided to use the PChi-C methodology. It is important to highlight that Hi-ChIP and similar methods are frequently used to target histone modifications, but, with few exceptions such as CTCF, the study of transcription factors is not currently compatible with this methodology. Besides, under our understanding, no p53 Hi-ChIP experiment has been reported so far.

One caveat of our approach could be related with the fact that p53 binding events were retrieved from a publicly available p53 ChIP-seq dataset generated using the same cellular model and drug concentration, but upon 12 hours of exposure. However, this is not a real limitation since it has been demonstrated that p53 binding is stimulus and tissue independent (Verfallie et al.;

Hafner et al.; Sullivan et al.) as: i) p53 has a strong affinity with its sequence-specific DNA motif comprising two decameric half-sites of the consensus sequence RRRCWWGYYY (R = A/G, W = A/T, Y = C/T), which mainly determines its bindings; ii) it does not relies on auxiliar factors; iii) it overrides nucleosome positioning and chromatin landscapes, acting as pioneer transcription factor. Therefore, p53-bound sites retrieved at 12 hours of Nutlin-3a treatments are a reliable proxy to determine p53 binding events at 1 or 10 hours. However, it is true that not all p53 binding events are functional. As opposed to the p53 binding events, their functionality does in fact display clear cell type- and stimulus- specificity. For this reason, we have generated H3K27ac ChIP-seq data using the same cell culture as those used for PCHi-C (*i.e.*, 0, 1 and 10 hours of Nutlin-3a treatment), to identify the set of functional p53 bindings events in each time point and condition.

Finally, we are aware that by using PCHi-C and p53 ChIP-seq, as opposed to Hi-ChIP, we cannot claim that p53 is physically bound to the enhancer in close physical proximity with a promoter via DNA looping. However, it is unlikely that p53 binding and DNA looping are mutually exclusive events. Indeed, our study supports the dependence between the presence of p53 and DNA loops rather than mutual exclusion, since: i) gain of activity at p53-bound enhancers upon p53 activation is associated with the gain of promoter activity and transcriptional upregulation of the looping gene (Fig. 5 E-K); and ii) looping erasure via RAD21 degradation hinders the broadcasting of activation from p53-bound enhancers to their distal target genes, reflecting p53-driven gene transcriptional upregulation (Fig. 7 D-F). Besides, following a suggestion from reviewer #3, to obstruct p53 binding we have deleted, via CRISPR-based methodology, the p53 sequence-specific recognition motif at two selected p53-bound enhancers. As showed in Fig. 7I, distal target genes are no longer upregulated in response to Nutlin-3a. Collectively, these results demonstrate that DNA loops connecting p53-bound enhancers and promoters of target genes are key for the upregulation of the engaged distal target gene upon cellular stress, mechanically coupling p53 binding and DNA loops.

Reviewer #3 (Remarks to the Author):

Serra et al. investigate changes in 3D genome conformation in a colon cancer cell line in which the p53 stress response was pharmacologically activated. The authors generated several omics datasets across a time series, from which they draw associations aimed at describing the relationship between p53 pathway-activated transcriptional changes and 3D genome conformation. Technically, the presented data appear of high quality and suitable for answering the research question at hand. Analyses largely appear appropriate, although at occasions superficial. However, I feel that the data is often overinterpreted and conclusions drawn too strong, especially considering a general lack of functional experiments to validate new p53 target genes/enhancers or no assessment of 3D genome conformation in Cohesin-depleted cells in which the p53 pathway was stimulated. As such, I think the paper falls short in delivering what is promised in the manuscript's rather boldly written abstract.

We thank the reviewer for highlighting the technical quality of the projects, as well as for the constructive suggestions to improve it. We completely agree that functional validation of new p53 target genes/enhancers, as well as the assessment of DNA loop dynamics in cohesin-depleted cells are fundamental to make strong claims. For this reason, we started both experiments before submitting the manuscript. However, since generating this data requires an extended amount of time, we were not able to include it in the first submission of the manuscript.

In the revised version of the manuscript, you will find: i) a function validation of two selected candidate genes based on CRISPR-based deletion of functional p53-bound enhancers, including two independent deletions in homozygosity for one p53-bound enhancer and two clones of a third deletion at a second p53-bound enhancer; and ii) experimental validation of DNA loops' erasure upon RAD21 depletion applying PChi-C methodology. Besides, we have explored the cell type-specificity of the set of new p53 target genes.

Specific comments:

1. I feel that the authors should clearly mention in their Introduction that there have been reports

linking p53 binding at enhancers to changes in 3D genome folding. Some of these papers are cited (i.e. refs 20-22), but deserve to be briefly discussed.

Thank you for the suggestion. We have now included extended information describing previous findings in the introduction of the manuscript.

2. The data in Supp.Fig.1A-C contains relevant QC information that is currently not used in the paper. The authors should discuss these data.

We have described data from Suppl. Fig. 1A-C in the results section of the manuscript. Moreover, following a suggestion of reviewer #2, we have extended this analysis. In the new version of the manuscript QC information is summarized in Suppl. Fig. 1A-G.

3. Fig.2F: I'm not sure how the authors can conclude from this graph that 'less insulating borders are preferentially erased during p53 activation'. This would require separately showing insulation scores at lost borders vs. stable (or gained) borders.

To demonstrate this claim, we have separated TAD borders into two categories: i) TAD borders erased along p53 activation; and ii) invariant TAD borders (*i.e.*, maintained or stable). Sigmoid insulation score profiles stacked over a 1Mb genomic region centered at TAD borders (Suppl. Fig. 2D) demonstrate that the less insulating borders are preferentially erased during p53 activation.

4. The authors use previously generated p53 ChIP-Seq data, although timepoints do not entirely match between studies it seems (e.g. 10h vs 12h)? The authors focus quite some of their analyses on their 10h time point, but does that mean that they used 12h p53 binding data? If so, this needs to be clearly stated.

One caveat of our approach could be related to the fact that p53 binding events were retrieved from a publicly available p53 ChIP-seq dataset generated on the same cellular model and drug concentration, but upon 12 hours of exposure. However, this is not a real limitation since it has

been demonstrated that p53 binding is stimulus and tissue independent (Verfallie et al.; Hafner et al.; Sullivan et al.). The underlying reason are the followings: i) p53 has a strong affinity with its sequence-specific DNA motive comprising two decameric half-sites of the consensus sequence RRRCWWGYYY (R = A/G, W = A/T, Y = C/T), which mainly determines its bindings; ii) p53 does not relies on auxiliar factors; iii) p53 overrides the nucleosome positioning and the chromatin landscapes, acting as pioneer transcription factor. Therefore, p53-bound sites retrieved at 12 hours of Nutlin-3a treatments are a reliable proxy to determine p53 binding events at 1 or 10 hours. However, it is true that not all p53 binding events are functional. As opposed to the p53 binding events, their functionality does in fact display clear cell type- and stimulus- specificity. For this reason, we have generated H3K27ac ChIP-seq data using the same cell culture as those used for PChi-C (*i.e.*, 0, 1 and 10 hours of Nutlin-3a treatment), to identify the set of functional p53 bindings events in each time point and condition.

We have clearly stated it in the manuscript, and we have provided further explanation about the underlying reasons that support the adequacy of our approach in the discussion.

5. I find Fig.3D-E very difficult to read and understand; I strongly encourage the authors to use a simpler way of visualising the enrichment of p53 binding in the different switching categories.

We agree that Fig. 3D-E displays similar data as in Fig. 3F-G (now Fig. 3C-D in the revised version of the manuscript), the former ones displaying the absolute values and the latter ones just the differential values for each pair-wise comparison. Furthermore, enrichment of p53 binding in the different switching categories is represented in Fig. 3C (now Fig. 3B). Due to this partial redundancy and the figure's complexity which hinders its understanding, we have decided to remove Fig. 3D-E following your suggestion.

6. Fig.2/3: why don't the authors attempt to connect TAD dynamics to transcriptional changes induced by p53 pathway activation, or p53 binding dynamics? This seems a very relevant angle to pursue.

Thanks for the suggestion. We previously explored the association between TAD border dynamics and p53 binding dynamics. As displayed in **Fig. H** and **I**, TAD borders associated with functional p53 binding sites (*i.e.*, TAD borders located up to 500kb away from a functional p53 binding site, $p53 < 500\text{kb}$) had similar insulation score profiles compared to TAD borders found at least 500kb away from any functional p53 binding site ($p53 > 500\text{kb}$).

Fig. H Box plots displaying delta scores (*i.e.*, derivative of the insulation score metric) of TAD borders associated (< 500kb) and not associated (> 500kb) with functional p53 sites after 1 hour (left) and 10 hours (right) of p53 activation.

Fig. I Sigmoid insulation score profiles stacked over a 1 Mb genomic region centered at TAD borders upon 1 hour (green) and 10 hours (purple) of p53 activation. TAD borders were classified as either associated with functional p53 sites (*i.e.*, TAD borders located up to 500Kb away from a functional p53 sites, $< 500\text{kb}$), displayed with a dotted line, or not associated with p53 ($> 500\text{kb}$), displayed with a solid line.

Then, we focused on DNA loops engaging enhancers and promoters with or without functional p53 binding sites at the enhancer. Comparing cells along p53 activation, we observed that TAD borders found between promoters and enhancers forming new interactions at 10h, compared to 1h of activation, are over 3 times more likely to be lost if there is a functional p53 bound to the enhancer (OR: 3.2, p value= $2e-9$; [[51, 98], [532, 3253]]) (**Fig. J**). However, we did not find

a significant association when comparing data from cells before and after 1 hour of p53 activation (OR: 0.95, p value= 7.50e-01; [[70, 96], [1747, 2272]]) or before p53 activation and 10 hour of Nutlin-3a treatment (OR: 01.01, p value= 9.34e-01; [[55, 116], [1544, 3301]]). For these reasons, we do not feel confident including these results in the manuscript.

Fig. J. Schematic representation of the odd ratio calculation to associate DNA looping between p53-bound enhancers and promoters and loss of TAD borders.

Finally, we have used our RNA-seq data to analyze expression changes of genes in the vicinity (closer than 250kb from the TAD border) of stable, gained or lost TAD borders associated or not associated with functional p53 binding sites throughout p53 activation. Overall, as displayed in **Fig. K**, genes associated with lost TAD borders tended to increase their expression levels, whilst genes associated with gained TAD borders tended to decrease their mRNA levels during p53 activation. However, p53's proximity to these TAD borders does not seem to influence these tendencies. In conclusion, we cannot claim an association between TAD dynamics and transcriptional changes induced by p53 pathway activation.

Fig. K. Box plots displaying gene expression changes (log₂ fold change) of genes in the vicinity (less than 250kb away from TAD border) of stable, gained or lost TAD borders associated or not associated with functional p53 binding during p53 activation.

7-. Line 198: ‘solely’ is a very strong word, the authors cannot exclude that examples exist where p53 represses or induced A->B switching, so I suggest using ‘mainly’ instead.

We have replaced the word “solely” by “mainly” in this phrase.

8. Fig.5G shows no transcriptional changes in the p53 target genes identified after 1h of activation, despite increases in p53 response element H3K27Ac levels. It seems at odds with p53-mediated topological/epigenomic changes being relevant for the acute transcriptional reprogramming induced by p53 pathway activation. In panel K, the authors show nascent RNA levels for some genes increasing at 1h, but in my opinion one still has to conclude that the majority of the immediate p53 target genes are not upregulated after 1h.

Using RNA-seq data, which measures steady state levels of mature RNAs (mRNAs), we did not observe transcriptional changes of genes controlled by p53 binding their promoters (named in Fig. 5B, G as “proximal”) and enhancers (named in Fig. 5B, G as “distal”) after 1 hour of Nutlin-3a treatment. However, genes from both categories were significantly activated according to H3K27ac gain at their promoters. Furthermore, quantification of nascent transcript levels of selected distal target genes (genes targeted by p53-bound enhancers) demonstrated a global and reproducible tendency of gene upregulation after 1 hour of p53 activation (Fig. 5K). It is true that only 4 out of the 10 genes analyzed showed significant differences, possibly due to limited sample size (*i.e.*, three independents *in vitro* p53 activation experiments were used for this study), and the experimental variability associated with *in vitro* p53 activation.

To test this hypothesis, we have increased the sample size by including extra biological replicates. As shown in the new Fig. 5K, after increasing the sample size we detect statistically significant transcriptional upregulation at the nascent transcript level of 11 out of 12 tested genes*. Collectively, this data, combined with the H3K27ac data (Fig. 5F), supports the coupling between topological, epigenetic, and transcriptional changes for the acute transcriptional reprogramming induced by p53 activation and suggests a limitation of RNA-seq data to capture these transcriptional changes.

* Please, note that we have analysed 12 instead of 10 genes because we have included two extra genes (*S100A1* and *PCMI*) that have been used for CRISPR-based validation (see comment 14) (Fig. 5K).

Regarding the RNA-seq data, there are two reasons that may explain its limitation in detecting transcriptional upregulation after 1 hour of p53 activation:

- As opposed to after 10 hours of Nutlin-3a treatment, a 1 hour treatment generally triggers lower activation, both according to H3K27ac levels (at p53-bound promoters, p53-bound enhancers and promoters of genes distally regulated by p53-bound enhancers) (Fig. 5 E-J) and nascent transcriptional output (Fig. 5K). Besides, this is also supported by the limited activation and accumulation of the p53 protein (Suppl. Fig. 1D-G) and the moderate upregulation of well-known p53 target genes (*e.g.*, *CDKN1A/p21*, *TGFA*, *BAX*) (Suppl. Fig. 1B, F, G)

- p53 activation leads to transcriptional changes of miRNAs and RNA binding proteins (RBPs) after 1 hour of its activation. Indeed, p53 has been identified as a major direct regulator of miRNAs (Feng et al.). Consequently, levels of mRNA (detected by RNA-seq) may not capture the slight increase of transcriptional rates detected by qRT-PCR analysis, since miRNAs and RBPs may affect mRNA stability and ultimately alter the steady state levels of mature RNAs. Specifically, 7 and 4 microRNAs and 84 and 36 RBPs are significantly upregulated and downregulated, respectively, after 1 hour of p53 activation (Fig. L). Among these, 2 microRNAs (miR-1293 and miR-1915) and 9 RBPs (*SRSF1*, *EIF3*, *BRBM4*, *RBM14*, *ESRP2*, *RBM12B*, *PUF60*, *DNAJC17* and *EIF3B*) are distally controlled by p53-bound enhancers. Moreover, miR-4738 has a functional p53 binding at its promoter region.

Fig. L Differential expression analysis between 1 hour (Nut 1h) and 0 hour (Nut 0h) time points. Differentially expressed genes are those with log2 fold change (\log_2FC) $\neq 0$ and adjusted p value ≤ 0.05 (horizontal dotted line). Differentially expressed microRNAs (MIR) and RNA binding proteins (RBP) are highlighted in green or blue, respectively.

Furthermore, the limited capacity to identify early p53-driven transcriptional response quantifying steady state levels of mRNA was previously described by Allen et al.. Specifically, by performing comparative analysis of nascent RNA levels by GRO-seq (Core et al.) vs. RNA steady state levels by microarray, they showed that many p53 target genes transcribed at low levels are missed by microarray. For instance, *CDKN1A* and *TP53I3* mRNAs were not significantly upregulated after 1 hour of 10 μ M Nutlin-3a treatment, but their nascent transcription levels were found significantly increased. Interestingly, the GRO-seq data generated treating HCT116 cells with 10 μ M Nutlin-3a treatment identified 5 miRNAs directly transactivated by p53 (miR-1204, miR-3189, miR-34a, miR4679-1 and miR-4692). Besides, it is worth highlighting that nascent transcripts represent around 0.5% of total RNA content in a cell.

Finally, we also compared this publicly available GRO-seq data with our RNA-seq data at 1h and 10h after Nutlin-3a treatment. In fact, we identify only 40/198 (20%) to be significantly upregulated in our RNA-seq data after 1h hour of treatment, whereas 144/198 (73%,) of the genes identified by GRO-seq are found significantly upregulated in our RNA-seq data 10 hours after treatment (**Fig. M**). Furthermore, out of the 40 genes identified at 1 hour, 39 are also identified at 10 hours. Altogether, this clearly reflects the fact that RNA-seq cannot capture the complete picture of the transcriptional effects occurring only 1 hour after p53 activation, given that this technique is used to measure the global mRNA abundance, which is most likely affected by the regulation of splicing events.

Fig. M Upset plot depicting the overlap between genes found upregulated in publicly available GRO-seq data (which measured nascent RNA levels in HCT116 cells after 30 minutes of p53 activation by Nutlin-3a), and genes found significantly upregulated by RNA-seq after 1 and 10 hours of p53 activation by Nutlin-3a in our data, named as DEG_Nut1h and DEG_Nut10h, respectively.

Summarizing, our data, supported by the literature, demonstrates that the genes that we identified as distal targets of p53-bound enhancers are globally upregulated after 1 hour of p53 activation, albeit at low levels, and these cannot be detected by measuring mRNA abundance.

9. Fig.6. How do these two different scenarios (*i.e.*, pre-established *vs de novo*) relate to the transcriptional changes at these target genes? This is important to investigate, as the authors claim that p53-mediated topological changes are underlying the transcriptional changes observed upon Nutlin-3a treatment.

Thank you for this suggestion. To compare the transcriptional response in the two different scenarios (pre-established *vs. de novo* interactions between p53-bound enhancers and target genes), we have: i) compared the H3K27ac levels (log₂FC) at the promoters of these target genes; and ii) compared the nascent expression levels (log₂FC) of these target genes as measured by qRT-PCR. This approach has been taken to compare both 1 and 10 hours timepoints to basal conditions (0h).

When comparing H3K27ac levels, we observed that, after 1 hour of Nutlin-3a treatment, target genes interacting with a p53-bound enhancer through a pre-established interaction gain significantly more H3K27ac at their promoter than those involved in *de novo* interactions (**Fig. N**). This observation was no longer hold at 10 hours of p53 activation, most likely due to the large window of time (*i.e.*, 10 hours instead of 1 hour), making it a lot more complicated to associate the dependency between the interaction's dynamism and any difference in transcriptional response.

Fig N. Differential H3K27ac between Nut 1h and Nut 0h (left) and Nut 10h and Nut 0h (right) time points (log₂FC) at promoters of genes in physical proximity with p53-bound enhancers. Genes were subdivided into two categories: i) genes engaged in *de novo* interactions with p53-bound enhancers (left, labelled as gained ints.); and ii) genes engaged in pre-established interactions with p53-bound enhancers (right, labelled as maint. ints.). Mann-Whitney U-Statistic was used to test whether the mean H3K27ac log₂FC differed significantly between groups.

Furthermore, these observations made using H3K27ac were also seen at the nascent transcript levels (**Fig. O**). Although not statistically significant, possibly due to few data points (only 10 genes were analyzed), the nascent expression levels of target genes involved in pre-established interactions tends to increase more than that of genes involved in *de novo* interactions.

Fig. O Distribution of nascent mRNA expression levels of distal target genes along p53 activation between Nut 1h and Nut 0h (left) and Nut 10h and Nut 0h (right) time points (log₂FC). A two-side Wilcoxon test was used to test whether expression levels differed significantly between conditions for each time point.

Collectively, this data suggests that pre-established loops do allow the effect of p53 activation to be broadcasted faster from enhancers to gene promoters than *de novo* loops. However, as this effect does not reach statistical significance in all comparisons, we do not feel confident to claim it in the manuscript.

10. Fig.7D. Most p53 target genes do not seem much affected by the Cohesin depletion, especially at the 1h timepoint (2 out of 10). This doesn't seem like a firm basis for concluding that 'Rad21 degradation compromises p53-driven transcription responses'.

We agree that nascent transcript levels are not always significantly affected by RAD21 depletion, especially after 1 hour of p53 activation, most likely due to the limited sample size (only three biological replicates, *i.e.*, three independent *in vitro* p53 activation experiments) and the experimental related variability during *in vitro* p53 activation. However, there is a clear and reproducible tendency of impairment of transcriptional upregulation in all cases (Fig. 7E-F; Suppl. Fig. 6C and Fig. 7D of the old version of the manuscript), whether statistically significant or not. Besides, this finding is supported by the significant reduction of H3K27ac gain at the promoter of the genes distally controlled by p53-bound enhancers (Fig. 7D; Fig. 7C of the old version of the manuscript).

To demonstrate a more significant effect of cohesin depletion on gene transcription we have increased the sample size by adding additional biological replicates. As shown on the Fig. 7F (Suppl. Fig. 6C of the old version of the manuscript), after increasing the sample size we are able to detect significant transcriptional upregulation at the nascent transcript level all 12 tested genes. Collectively, this data, in combination with H3K27ac data (Fig. 7D; Fig. 7C of the old

version of the manuscript), demonstrate that RAD21 degradation globally reduces p53-driven transcription responses.

11. Data presented in Fig.S6C seems to show a more convincing transcriptional impact of Rad21 depletion on p53 target genes. How was this data generated (qPCR as well)? And why is it different from the data shown in 7D?

Both figures represent the quantity of nascent transcripts of distally regulated genes by p53-bound enhancers measured by qRT-PCR. The main difference lies in the normalization procedure.

Fig. 7D (Fig. 7F of the new version of the manuscript) represents relative nascent mRNA expression (fold change). Specifically, nascent transcript expression fold change was calculated using the $\Delta\Delta C_t$ method (Livak et al.,) against *HPRT1* as the housekeeping gene. We adjusted a linear mixed model to relative nascent mRNA expression to decide whether gene expression can be explained by the fixed-effect of the experimental condition (either the effect of Nutlin-3a over time or degradation of RAD21), or by a random effect. p-values correspond to the rejection of a hypothesis which states that the random effect fits better than the experimental condition.

Suppl. Fig. 6C (Fig. 7E of the new version of the manuscript) displays the distribution of wild-type-normalized nascent mRNA expression levels (fold change). Nascent mRNA expression at each timepoint was divided by the average expression of genes in WT conditions (which is why the median of WT conditions are always precisely 1). A two-side Wilcoxon test was used to test whether expression levels differed significantly between conditions for each time point (star). To clarify this, we have extended the description of Fig. 7E.

12. The lack of (PC)Hi-C data on the Rad21-depleted nutlin-treated cells precludes any conclusions on whether p53-mediated topological changes are cohesin-dependent, such as those stated in the abstract: ‘Finally, we showcase that p53 controls transcription of distal genes through newly formed and pre-existing enhancer-promoter loops in a cohesin dependent manner.’

Thank you for the constructive suggestion. In the manuscript we have included a PChi-C dataset generated after treating two biological replicates of HCT116 cells with Nutlin-3a during 1 hour in RAD21-depleted conditions (*i.e.*, 6 hours treatment with 1 μ M of 5-Ph-IAA followed by 1 hour treatment with 10 μ M of Nutlin-3a).

Our observations do back up the conclusions obtained, being that most p53-mediated topological changes act in a cohesin-dependent manner (see new Fig. 7A-C). Similarly to previously reported findings (Thiecke *et al.*), cohesin depletion leads to a dramatic loss of promoter interactions. Specifically, the mean CHiCAGO score of the 179 interactions engaging p53-bound enhancers at 1 hour of Nutlin-3a activation in RAD21-depleted cells is significantly lower compared to wild-type conditions (Wilcoxon $p < 2.2e-16$). 67% of the DNA loops engaging p53-bound enhancers and candidate target genes are completely erased (*i.e.*, CHiCAGO score < 5) and 76% significantly reduce their interaction strength upon cohesin depletion. As previously reported (Rinzema *et al.*), cohesin is mechanistically involved specifically with long-range interactions. In fact, the mean genomic distance of the interactions lost in RAD21-depleted cells is significantly higher (171kb) compared to the stable interactions (24kb). In conclusion, we believe that these results showcase that p53 controls transcription of distal genes through newly formed and pre-existing enhancer-promoter loops in a cohesin dependent manner.

13. Overall, the authors state that one of the key strengths of this paper is that it provides a reliable set of new p53 target genes. However, these genes might be very specific to the colon carcinoma cell line used. How do the 340 supposedly direct target genes - which the authors only based on a measured PChi-C interaction (see comment below) - hold up in other cell systems in which the p53 pathway is activated?

To investigate the cell type-specificity of the core transcriptional program of genes directly regulated by p53-bound enhancer over distance we have treated with 10 μ M of Nutlin-3a: i) one hepatocellular carcinoma (Hep G2), one neuroblastoma (SH-SY5Y), one adenocarcinoma (MCF7), one melanoma (C32) and one clear cell renal cell carcinoma (CAKI-1), all of these human cancer cell lines wild-type for p53. Besides, we also treated 3 types of primary human cell types: i) umbilical vein endothelial cells (HUVEC); ii) dermal lymphatic endothelial cells

(HDLEC); and iii) pericytes. Cells were collected after 1 and 10 hours of treatment. Besides, we also treated cells with the drug vehicle (*i.e.*, DMSO) and used these cells as a control of basal conditions without p53 activation (also referred to as 0 hours of Nutlin-3a). Nascent transcript levels of 3 well-known p53 target genes (*i.e.*, *CDKN1A*, *TGFA*, *BAX*) and 12 target genes distally regulated by p53-bound enhancers were analyzed by qRT-PCR in each cell line or primary cell type and compared with data previously generated in HCT116 cells (Suppl. Fig. 6 and 7).

Globally, the p53-driven transcriptional response is conserved across cell types, but the kinetics and strength of the response varies between cell types and genes. For instance, the transcriptional response is a little bit delayed in MFC7 and Hep G2 cells. C32 cells do not follow the general tendency of progressive gene upregulation over time compared with the other cell types. Besides, JAG2 is more upregulated upon p53 activation in pericytes than HDLEC.

In summary, although a deeper study is required to make solid conclusions about the cell type-specific response to p53 activation, study which may well lead to the generation of another manuscript, our results support the reliability of a new set of p53 target genes that we have identified and the partial conservation across cell types of the p53 transcriptional response.

14. I find the evidence presented for ‘a reliable set of new p53 direct target genes’ (quote from their abstract) rather underwhelming. Sure, these genes are linked in 3D to a p53-bound enhancer, but that is no formal proof that these enhancers actually regulate these genes and that p53 is critical for the enhancing effect of said element. Since the authors use a cell line model that is amenable to genome editing (as shown by Fig.7), I would have expected the authors to use CRISPR-Cas9 technology to experimentally validate at least a few of their hypothesised enhancer-promoter links by cutting out the p52 enhancer - or even better disrupt the p53 binding motif in the enhancer. It is this type of proof that the authors in my opinion need to make a compelling case for a novel set of reliable p53 target genes - especially if as stated in the discussion most of these new target genes have not been detected by the dozens of earlier studies conducted.

We agree that functional validation of p53-bound enhancer-gene associations is the more accurate and definitive proof that the new set of genes are distally regulated by p53. For this reason, we have selected two target genes (*i.e.*, *S100A1* and *PCMI*) controlled by a single p53-bound enhancers located 101kb and 202Kb from the promoter region respectively (Fig. 7G-H). Using CRIPR-Cas9 methodology, we have generated homozygotic clones lacking the p53 sequence-specific DNA motive comprising two decameric half-sites of the consensus sequence RRRCWWGYYY (R = A/G, W = A/T, Y = C/T) (Suppl. Fig. 9G). Cellular clones were treated with 10 μ M of Nutlin-3a and were collected after 1 hour, as well as with the drug vehicle (*i.e.*, DMSO). Wild-type HCT116 cells were used as control. After validation of p53 activation by qRT-PCR and WB (Suppl. Fig. 9H-I), we quantified nascent transcription of the corresponding distal target gene associated with the deleted p53 binding site. As shown in Fig. 7I, both distal target genes show hindered transcriptional upregulation upon p53 activation after deletion of the p53 sequence-specific DNA motif.

Collectively, these results support reliability of the new set of p53 direct target genes distally controlled by p53-bound enhancers and uphold the power of integrating 3D genome architecture, epigenetics, gene expression and transcription factor binding profiles to discover new transcription factor target genes.

REVIEWERS' COMMENTS

Reviewer #1 (Remarks to the Author):

First, I would like to congratulate Serra et al on their comprehensive revisions. The new experiments add significantly to the narrative and conclusions of the work. I have a few further suggestions:

There are still elements of the manuscript that need grammatical and linguistic polish. Line 33 -34, Line 137, Line 416, among others.

More concerningly, there are still examples of inaccurate scientific language within the manuscript eg Line 119 "most compartments" = how many is most? 99%? 51%? Line 165 – "significant proportion".

I still believe that vastly more figure panels are shown in primary figures than are required to communicate the authors findings. In fact, I believe sifting through these panels detracts from the clarity of the authors message. As the authors suggest, we should leave this to the Editors discretion.

I would like to see more discussion of the interesting reversal of genome architecture upon Nutlin-3a wash eg why is wave 2 more predominantly reversed than wave 1?

Reviewer #2 (Remarks to the Author):

The revised manuscript satisfies my criticisms and includes a significant amount of new data (including more functional validation). Overall, I am happy with the current version and I have no further comment. Congratulation to the authors for the work and revisions.

Reviewer #3 (Remarks to the Author):

The authors have done the necessary and much needed experiments to now sufficiently support their original claims. I want to commend them for following up on my - and also rev#1's - criticisms with solid experiments (in particular the CRISPR-Cas9 work), which has made the paper much stronger. I now support publication.

Point-by-point responses to Reviewers' comments (NCOMMS-23-26694-T).

First, we would like to thank the reviewers for the constructive criticism and suggestions to improve the manuscript. To help with the structuring of this document, we have marked the reviewer's comments with **green typography**, while our replies are shown in normal black font.

Reviewer #1 (Remarks to the Author):

First, I would like to congratulate Serra et al on their comprehensive revisions. The new experiments add significantly to the narrative and conclusions of the work. I have a few further suggestions:

There are still elements of the manuscript that need grammatical and linguistic polish. Line 33-34, Line 137, Line 416, among others. More concerningly, there are still examples of inaccurate scientific language within the manuscript *e.g.*, Line 119 “most compartments” = how many is most? 99%? 51%? Line 165 – “significant proportion”. -I still believe that vastly more figure panels are shown in primary figures than are required to communicate the authors findings. In fact, I believe sifting through these panels detracts from the clarity of the authors message. As the authors suggest, we should leave this to the Editors discretion. I would like to see more discussion of the interesting reversal of genome architecture upon Nutlin-3a wash eg why is wave 2 more predominantly reversed than wave 1?

We thank the reviewer for the constructive suggestions to improve the manuscript. All suggestions, including grammatical and linguistic revision and more precise descriptions, have been addressed. Besides, we have provided a more detailed discussion regarding the reversal of genome architecture upon p53 inactivation.

Reviewer #2 (Remarks to the Author):

The revised manuscript satisfies my criticisms and includes a significant amount of new data (including more functional validation). Overall, I am happy with the current version and I have no further comment. Congratulation to the authors for the work and revisions.

We thank the reviewer for the constructive suggestions to increase the quality of the manuscript.

Reviewer #3 (Remarks to the Author):

The authors have done the necessary and much needed experiments to now sufficiently support their original claims. I want to commend them for following up on my - and also rev#1's - criticisms with solid experiments (in particular the CRISPR-Cas9 work), which has made the paper much stronger. I now support publication.

We thank the reviewer for their time and constructive feedback to improve the manuscript.